# The novel lncRNA *lnc-NR2F1* is pro-neurogenic and mutated in human neurodevelopmental disorders

Cheen Euong Ang[1,2†], Qing Ma[3,4,5†], Orly L Wapinski[3,4,5†], ShengHua Fan[6], Ryan A Flynn[3,4,5], Qian Yi Lee[1,2], Bradley Coe[7], Masahiro Onoguchi[3,4,5], Victor Hipolito Olmos[1], Brian T Do[3], Lynn Dukes-Rimsky[6], Jin Xu[3], Koji Tanabe[1], LiangJiang Wang[8], Ulrich Elling[9], Josef M Penninger[9], Yang Zhao[3], Kun Qu[3,9§], Evan E Eichler[7], Anand Srivastava[6,8‡], Marius Wernig[1‡*], Howard Y Chang[3‡*]

[1]Department of Pathology, Institute for Stem Cell Biology and Regenerative Medicine, Stanford University, Stanford, United States; [2]Department of Bioengineering, Stanford University, Stanford, United States; [3]Center for Personal Dynamic Regulomes, Stanford University, Stanford, United States; [4]Department of Dermatology, Stanford University, Stanford, United States; [5]Department of Genetics, Stanford University, Stanford, United States; [6]JC Self Research Institute of Human Genetics, Greenwood Genetic Center, Greenwood, United States; [7]Department of Genome Sciences, Howard Hughes Medical Institute, University of Washington, Seattle, United States; [8]Department of Genetics and Biochemistry, Clemson University, Clemson, United States; [9]Institute of Molecular Biotechnology of the Austrian Academy of Science (IMBA), Vienna Biocenter, Vienna, Austria

**\*For correspondence:**
wernig@stanford.edu (MW);
howchang@stanford.edu (HYC)

[†]These authors contributed equally to this work
[‡]These authors also contributed equally to this work

**Present address:** [§]CAS Key Laboratory of Innate Immunity and Chronic Diseases, School of Life Sciences and Medical Center, University of Science and Technology of China, Hefei, China

**Competing interests:** The authors declare that no competing interests exist.

**Abstract** Long noncoding RNAs (lncRNAs) have been shown to act as important cell biological regulators including cell fate decisions but are often ignored in human genetics. Combining differential lncRNA expression during neuronal lineage induction with copy number variation morbidity maps of a cohort of children with autism spectrum disorder/intellectual disability versus healthy controls revealed focal genomic mutations affecting several lncRNA candidate loci. Here we find that a t(5:12) chromosomal translocation in a family manifesting neurodevelopmental symptoms disrupts specifically *lnc-NR2F1*. We further show that *lnc-NR2F1* is an evolutionarily conserved lncRNA functionally enhances induced neuronal cell maturation and directly occupies and regulates transcription of neuronal genes including autism-associated genes. Thus, integrating human genetics and functional testing in neuronal lineage induction is a promising approach for discovering candidate lncRNAs involved in neurodevelopmental diseases.
DOI: https://doi.org/10.7554/eLife.41770.001

## Introduction

Eukaryotic genomes are extensively transcribed to produce long non-coding RNAs (lncRNAs) in a temporally and spatially regulated manner (*Flynn and Chang, 2014*). Until recently, lncRNAs were often dismissed as lacking functional relevance. However, lncRNAs are emerging as critical regulators of diverse biological processes and have been increasingly associated with a wide range of diseases, based primarily on dysregulated expression (*Wapinski and Chang, 2011*). LncRNAs represent a new layer of complexity in the molecular architecture of the genome, and strategies to validate disease relevant lncRNAs are much needed. High-throughput analyses have shown that lncRNAs are widely expressed in the brain and may contribute to complex neurodevelopmental processes

(*Wapinski and Chang, 2011*; *Fertuzinhos et al., 2014*; *Valadkhan and Nilsen, 2010*; *Lv et al., 2013*; *Aprea et al., 2013*; *Ramos et al., 2013*; *Ramos et al., 2015*; *Ng et al., 2013*). However, few studies have examined the role of lncRNAs in brain development mostly due to technical difficulties. Direct lineage conversion by the transcription factors Brn2, Ascl1 and Myt1l (termed BAM factors in combination) into induced neuronal (iN) cells, recapitulates significant events controlling neurogenesis programs (*Vierbuchen et al., 2010*; *Wapinski et al., 2013*; *Ang and Wernig, 2014*), and therefore, it is a facile and informative system to study the role of lncRNAs in the establishment of neuronal identity.

The noncoding genome has emerged as a major source for human diversity and disease origins. Given that less than 2% of the genome encodes protein-coding genes, the majority of the genomic landscape is largely encompassed by non-coding elements. Efforts to identify genetic variation linked to human disease through genome-wide association studies revealed a significant majority affecting the non-coding landscape. Based on their expression and diversity in the mammalian brain, we postulate neuronal lncRNAs may be recurrently affected by mutations that disrupt normal brain function. Neurodevelopmental disorders manifest as a spectrum of phenotypes particularly early in life (*Voineagu et al., 2011*). Recent studies suggest that this diversity is the result of different combinations of mutations in multiple genes, often impacting key pathways such as synapse function and chromatin regulation. Nonetheless, despite recent findings that have greatly increased the number of protein coding genes implicated in human intellectual disability and autism, a majority of patients lack well-understood genetic lesions which include a large number of inherited variants occur in non-coding regions that could not be interpreted (*Iossifov et al., 2014*; *Ronemus et al., 2014*; *Gilman et al., 2011*; *Iossifov et al., 2012*; *De Rubeis et al., 2014*; *O'Roak et al., 2012a*; *O'Roak et al., 2012b*; *Hormozdiari et al., 2015*).

In this study, we used an integrative approach to identify lncRNA genes important for human disease by incorporating high throughput cell fate reprogramming, human genetics, and lncRNA functional analysis. In addition, we developed a pipeline to enrich for lncRNAs with neuronal function and are associated with disease through focal mutations in patients with autism spectrum disorder and intellectual disability (ASD/ID). Furthermore, we show that one of these lncRNAs, *lnc-NR2F1* participates in neuronal maturation programs in vitro by regulating the expression of a network of genes previously linked to human autism.

## Results

### LncRNA candidate loci are recurrently mutated in patients with neurodevelopmental disorders

LncRNAs have been associated with human diseases primarily through alterations in expression levels (*Meng et al., 2015*; *Cheetham et al., 2013*; *Gupta et al., 2010*). However, little is known about mutations affecting the genomic loci that encode lncRNAs. We previously profiled mouse embryonic fibroblasts (MEFs) expressing doxycycline-induced BAM factors after 48 hr, 13 and 22 days of expression (GSE43916). Surprisingly, annotation of the iN cell reprogramming transcriptome revealed that the majority of regulated transcripts were in fact non-coding elements (*Figure 1—figure supplement 1A*). Specifically, 58% of the changed transcripts corresponded to non-coding genes while only 42% of them corresponded to coding genes. About two thirds of these non-coding transcripts were composed of novel lncRNAs (*Figure 1—figure supplement 1B*).

To study the vast non-coding iN cell reprogramming transcriptome, we developed a rigorous pipeline to select lncRNAs with strong neuronal association (*Figure 1—figure supplement 1C*). We considered expression pattern during MEF-to-iN cell reprogramming and across mouse brain development, protein coding potential, chromatin enrichment, and 'guilt-by-association' with neuronal Gene Ontology (GO) terms. We observed 287 non-coding transcripts significantly changed expression during this time course (RPKM >1, fold change >2, p-value<0.05) (*Figure 1—figure supplement 1D*). Notably, lncRNAs that increased expression during early stages of iN cell reprogramming are more highly expressed in embryonic mouse brain, specifically in ventricular and subventricular zones where neurogenesis occurs; whereas lncRNAs that increased expression during intermediate to late stages of iN cell reprogramming were more highly expressed in adult mouse brain, including in mature cortical layers (*Figure 1—figure supplement 1D and E*). Furthermore, robust expression

of iN cell lncRNAs in the mouse brain confirmed that these transcripts are indeed bona fide neuronal transcripts.

We next assessed lncRNA association with chromatin, reasoning that such RNAs are more likely to exert gene regulatory function as non-coding RNAs. We performed histone H3 immunoprecipitation, followed by deep sequencing of associated RNAs (histone H3 RIP-seq), and discovered some of these 287 iN lncRNAs are chromatin-associated compared to IgG control and input in neural precursor cells (NPCs) or adult mouse brain tissue, suggesting that some of them may have more direct roles in gene expression control at the chromatin (*Figure 1—figure supplement 1F*). These 287 lncRNAs were then selected for further investigation. To further prioritize candidate lncRNAs, we determined regulatory modules based on patterns of co-expression between mRNAs and lncRNAs, inferring co-regulation from co-expression. Among the three predominant modules found, one was strongly associated with neuronal GO terms, such as *neurogenesis*, *axonogenesis*, and *synaptic organization and biogenesis* (*Figure 1—figure supplement 1G*). The remaining two modules consisted of broad non-neuronal biological functions. Based on the criteria included in the pipeline, we nominated 35 iN cell lncRNAs as most promising for possessing functions in the brain and confirmed their expression qRT-PCR (*Figure 1—figure supplement 1H*, *Figure 1—figure supplement 2A*). Collectively, these results suggest that MEF-to-iN cell reprogramming can be used to identify lncRNAs expressed in the developing and adult brain. (*Figure 1—figure supplement 1A–H*, *Figure 1—figure supplement 2*).

We next interrogated these 35 mouse lncRNA loci in patients with autism spectrum disorder and intellectual disability (ASD/ID). Firstly, we found that 28 of the 35 mouse lncRNA candidates have human synteny, and 10 loci were already annotated as non-coding RNAs (*Figure 1A* and *Figure 1—figure supplement 3A*). We next overlapped the 28 human lncRNA candidates and the remaining iN cell-lncRNAs coordinates to a CNV morbidity map recently built from 29,085 patients diagnosed with a spectrum of neurodevelopmental disorders and craniofacial congenital malformations, and 19,584 controls (*Coe et al., 2014*; *Cooper et al., 2011*). This approach was motivated by the fact that the CNV morbidity map has successfully identified novel syndromes characterized by recurrent mutations affecting protein-coding genes of ASD/ID patients, and has offered mechanistic insight into the drivers of the pathogenesis (*Coe et al., 2014*; *Cooper et al., 2011*; *Turner et al., 2017*).

Intersecting genomic coordinates of human lncRNA candidate loci to the CNV morbidity map revealed seven focal CNVs enriched in disease that overlap with five candidate lncRNA loci: E (FLJ42709), H (LOC339529), Z (LOC100630918), D (LINC00094) and O (LOC467979) (Sequences in supplementary documents). Among these seven focal CNVs, five events corresponded to small deletions in human lncRNA candidates E, H, Z, D and O. Two lncRNA loci corresponding to lncRNAs H and Z were affected by two independent and different focal CNVs. We verified that all five human lncRNAs are expressed during human brain development (*Figure 1—figure supplement 3A*). We then designed a custom tilling array with dense coverage of the affected loci for comparative genomic hybridization (CGH) to validate the focal CNVs in the genomic DNA from affected individuals. 5 of 7 focal CNVs affecting the lncRNA loci were tested and validated (*Figure 1A and B*, *Figure 1—figure supplement 4A*). We could not test the last two CNVs because patient DNA was no longer available.

One of the focally deleted lncRNA was NR_033883 (also known as or LOC339529). This lncRNA locus is disrupted by two focal CNVs in two distinct ASD/ID patients: 990914 and 9900589 (*Figure 1—figure supplement 4A*). The NR_033883 locus neighbors the coding genes *ZFP238* (also known as ZBTB18, ZNF238, and RP58) and *AKT3*. Because the human NR_033883 locus is most proximal but does not overlap *ZFP238*, we propose to refer to this lncRNA as *lnc-ZFP238*. Intriguingly, we previously identified ZFP238 as a key downstream target of the Ascl1 network during MEF to iN cell reprogramming (*Wapinski et al., 2013*). Additionally, ZFP238 has an important role in neuronal differentiation during brain development (*Xiang et al., 2012*; *Ohtaka-Maruyama et al., 2013*; *Baubet et al., 2012*), and thus, *lnc-ZFP238* could have a promising neurogenic role given its high expression pattern during the early stages of direct neuronal reprogramming, as well as in postnatal mouse and human brain (*Figure 1—figure supplement 2*).

Another focally deleted lncRNA was the locus harboring human FLJ42709 (or NR2F1-AS1) (*Figure 1B*) which is adjacent to the protein-coding gene *NR2F1* (also known as *COUP-TF1*), encoding a transcription factor involved in neurogenesis and patterning (*Ramos et al., 2013*; *Armentano et al., 2006*; *Borello et al., 2014*; *Faedo et al., 2008*; *Harrison-Uy et al., 2013*;

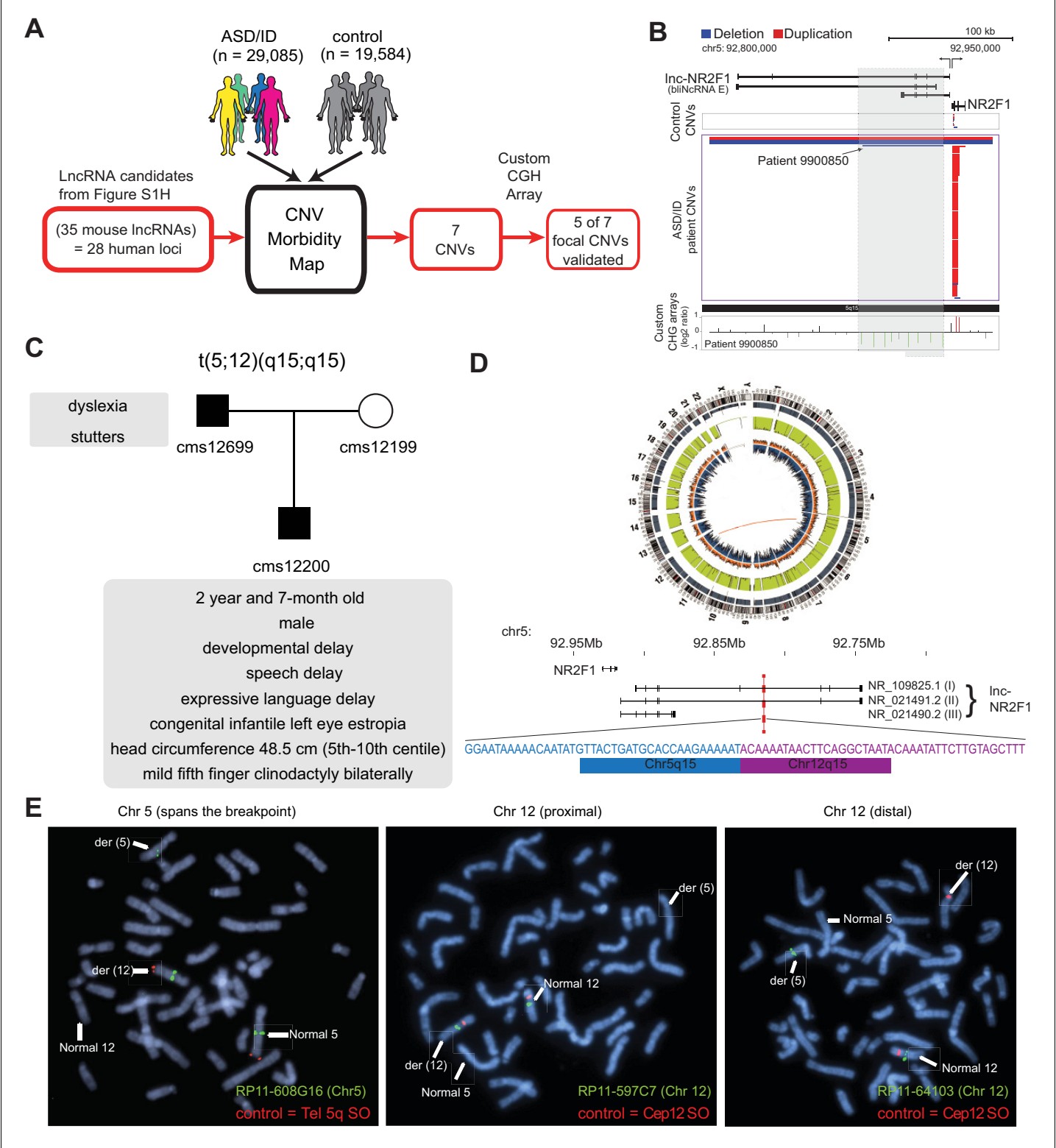

**Figure 1.** lncRNA loci are recurrently mutated in patients with neurodevelopmental disorders. (A) Schematic representation of CNV morbidity map analysis for candidate lncRNAs and all other iN lncRNAs loci. The 35 mouse lncRNA candidates (28 human loci) is from *Figure 1—figure supplement 1H*. (B) Top: Representative tracks for lncRNA E locus, also known as *lnc-NR2F1*. Depicted in blue are deletions and in red duplications. Arrow points to patient with focal deletion affecting the *lnc-NR2F1* locus only. Bottom: Custom CGH arrays used to validate chromosomal aberration in patient 9900850 harboring focal deletion represented in green signal. (C) Genetic pedigree analysis for family with paternally inherited balanced chromosomal

*Figure 1 continued on next page*

*Figure 1 continued*

translocation (5;12) (q15;q15), including a summary of clinical features for patient CMS12200 and father. The mother has a normal karyotype. Listed in the box are the symptoms of the patients. (D) Top: Circa plot representing the pathogenic chromosomal event for patient CMS12200 involving chromosomes 5 and 12. Bottom: Representative chromosome ideogram and track of the balanced chromosomal break affecting patient CMS12200. Below the ideoplot is the schematic representation of predominant human isoforms for *lnc-NR2F1* and the site of the break site disrupting the long isoforms. (E) The locations of the probes are in *Figure 1—figure supplement 4C*. Left: Metaphase spread from patient CMS12200 with the t(5;12) translocation showing FISH signals obtained with the clone RP11-608G16 (green) spanning Chromosome five breakpoint, and a Chromosome five telomere-specific probe (red). Middle: Metaphase spread from patient CMS12200 with the t(5;12) translocation showing FISH signals obtained with the clone RP11-597C7(green) proximal to Chromosome 12 breakpoint, and a Chromosome 12 centromere-specific probe (red). Right: Metaphase spread from patient CMS12200 with the t(5;12) translocation showing FISH signals obtained with the clone RP11-641O3 (green) distal to Chromosome 12 breakpoint, and a Chromosome 12 centromere-specific probe (red).

DOI: https://doi.org/10.7554/eLife.41770.002

The following figure supplements are available for figure 1:

**Figure supplement 1.** Molecular profiling of direct fibroblast to iN cell reprogramming nominates functional lncRNAs involved in neurogenesis.
DOI: https://doi.org/10.7554/eLife.41770.003

**Figure supplement 2.** QRT-PCR validation of candidate lncRNAs expression.
DOI: https://doi.org/10.7554/eLife.41770.004

**Figure supplement 3.** Conserved lncRNAs have distinct pattern of expression across different stages of the human brain.
DOI: https://doi.org/10.7554/eLife.41770.005

**Figure supplement 4.** Other reports of CNVs affecting *lnc-NR2F1* and an example of focal deletion affecting *lnc*-ZFP238 and characterization fo patient CMS12200.
DOI: https://doi.org/10.7554/eLife.41770.006

*Lin et al., 2011*; *Job and Tan, 2003*; *Tsai and Tsai, 1997*; *O'Leary et al., 2007*). This lncRNA was previously annotated as 'NR2F1-antisense 1' (NR2F1-AS1). However, our RNA-seq analysis showed that at least one isoform of the mouse lncRNA and all detected isoform of the human lncRNA are transcribed divergently from *NR2F1* without antisense overlap (*Ramos et al., 2013*). For scientific accuracy, we therefore propose the name *lnc-NR2F1*. We first asked whether the coding gene *NR2F1* could also be affected by the focal CNVs in ASD/ID patients. Detailed statistical analysis of the primary data taking into account the relative probe density suggested that inclusion of *NR2F1* is not statistically significant compared to the control group. Moreover, we precisely mapped the independent focal deletion found in patient 9900850 by CGH analysis and found only the *lnc-NR2F1* locus to be disrupted (*Figure 1B*). These results implicated the genetic disruption of *lnc-NR2F1* as likely contributor to complex neurodevelopmental disorders.

Chromosomal aberrations encompassing the *lnc-NR2F1* locus and additional genes on chromosome 5q14 have been previously reported in several patients with neurodevelopmental deficits and congenital abnormalities (*Figure 1—figure supplement 4B* and *Supplementary file 1*) (*Al-Kateb et al., 2013*; *Brown et al., 2009*; *Cardoso et al., 2009*; *Malan et al., 2006*). However, given that several genes are affected by the deletions, the particular contribution of each gene was difficult to resolve. Three patients (5-year-old girl, 5 and 7 year old boys) with a de novo deletion of chromosome 5q14.3–15 were diagnosed with epileptic episodes, intellectual disability, bilateral periventricular heterotopia in the temporal and occipital horns of the lateral ventricles, minor dysmorphic facial features, developmental delay, and impaired to negligible language skills (*Figure 1—figure supplement 4B* and *Supplementary file 1*). The shared minimal deleted region between the patients spans 5.8 Mb and encompasses several annotated genes, amongst them lnc-*NR2F1* (*Cardoso et al., 2009*). Clinical examination of one of the patients that harbors a finer 6.3 Mb interstitial deletion, showed macrocephaly (>98[th] centile) and brain MRI revealed polymicrogyria. No cortical abnormalities were detected on the brain MRI for the other two patients (*Supplementary file 1*). More recently, an 8 year-old and 3 month-old boy with a de novo 582 kb deletion was diagnosed with global developmental delay, dysmorphic features, visual motor integration deficit, visual perception disorder, mild conductive hearing loss, and severe fine motor skills abnormalities (*Al-Kateb et al., 2013*) (*Figure 1—figure supplement 4B* and *Supplementary file 1*). Head circumference was 8[th] centile. Brain MRI revealed bilateral optic nerve atrophy. The 582 kb deletion affects the genes *NR2F1*, lnc-*NR2F1*, *FAM172A*, *POU5F2*, and *MIR2277*. A four year-old girl with a balanced de novo paracentric chromosome five inversion, inv(5) (q15q33.2), and microdeletions near

the rearrangement breaking points completely removed *NR2F1* and lnc-*NR2F1* (*Figure 1—figure supplement 4B* and *Supplementary file 1*). Additional genes on different chromosomes are affected by microdeletions and could potentially contribute to the phenotype. The patient was diagnosed with syndromic deafness, feeding difficulties, dysmorphism, strabism, and developmental delay.

Across those patients with structural variation encompassing the *lnc-NR2F1* locus in the literature, the minimal deleted region is approximately 230 kb, a small area encompassing the genes *NR2F1* and *lnc-NR2F1* (*Figure 1—figure supplement 4B* and *Supplementary file 1*). The most notable overlapping phenotype consists of global developmental delay, facial dysmorphism, and hearing loss. Hypotonia and opththalmological abnormalities are also common diagnoses (*Al-Kateb et al., 2013*) (*Figure 1—figure supplement 4B* and *Supplementary file 1*). Phenotypic heterogeneity amongst patients could be the result of dosage sensitive genes, polymorphisms on the unaffected allele, genomic variability, gender, and age, amongst others.

Independent of the patients previously reported (*Al-Kateb et al., 2013*; *Brown et al., 2009*; *Cardoso et al., 2009*; *Malan et al., 2006*), we identified a paternally inherited balanced translocation t(5;12) (q15;q15) in a 2 year and 7-month-old male patient (CMS12200) (*Figure 1C–D* and *Figure 1—figure supplement 4B*). Patient CMS12200 was diagnosed with developmental delay, speech delay, significant expressive language delay, and congenital infantile left eye esotropia (*Figure 1C*). Physical examination revealed small head size (head circumference 48.5 cm; fifth-tenth centile), and mild fifth finger clinodactyly bilaterally. Other physical features were normal. The patient's father was diagnosed with dyslexia and stutters, and carried the identical t(5;12) translocation. The patient's mother had a normal 46, XX karyotype (*Figure 1C*). Fluorescence in situ hybridization and whole genome sequencing defined the chromosomal breakpoints with high precision and revealed that only the *lnc-NR2F1* gene is disrupted in this patient (*Figure 1D and E*, *Figure 1—figure supplement 4C*). In humans, three predominant isoforms of *lnc-NR2F1* have been detected in neuronal tissue. The long isoforms (1 and 2) are affected by the chromosomal break, while the gene structure of the short isoform (3) could remain unaffected based on the location of the break (*Figure 1D*). Further studies by Sanger sequencing of the 5q15 and 12q15 breakpoint-specific junction fragments showed the identical breakpoints in the patient's father and revealed a loss of 9 nucleotides at the 5q15 chromosome and a loss of 12 nucleotides at the 12q15 chromosome in the patient and his father (*Figure 1D*). The breakpoint at 12q15 occurred in a coding gene desert and did not disrupt any coding genes, and is predicted to destabilize the affected transcript due to loss of 3' splice or polyadenylation signals (*Figure 1D* and *Figure 1—figure supplement 4D–E*). Importantly, whole genome sequencing data indicated the absence of other deleterious mutations known to be associated with autism or intellectual disability, to our knowledge (*Figure 1D*). Also, the genes adjacent to the break point (*FAM172A, ARRDC3, KIAA0625, USP15*) are not significantly changed (*Figure 1—figure supplement 4E*). Given that *lnc-NR2F1* is the only disrupted gene in this patient family, it is possible that haploinsufficiency is the primary cause for this syndrome and contribute to the phenotypes manifested across patients mentioned above. Further studies including a larger sample size and independent cases are required to conclusively link lnc-*NR2F1* mutations to multiple clinical symptoms described above.

## Molecular and functional characterization of *lnc-Nr2f1*

Given its potential involvement in neurodevelopmental disease, we next sought to investigate the function of *lnc-Nr2f1*. We focused on mouse *lnc-Nr2f1* as experimental approaches are more tractable in mouse models given the availability of a plethora of genetic tools. Remarkably, and in contrast to many other lncRNAs, *lnc-Nr2f1* was not only syntenically conserved in its genomic context (*Figure 2—figure supplement 1A*, *Supplementary file 6*) (*Quinn et al., 2016*), but also highly sequence conserved among all human lnc-*NR2F1* isoforms and the three exons in mouse *lnc-Nr2f1* (*Figure 2A*, *Figure 2—figure supplement 1B*). In addition, we identified short stretches of sequence homology (termed microhomology) near the conserved exons (exon 2 and 3 of human lnc-*NR2F1)* across different species, with recurrent sequence motifs and motif order conserved across different species (*Figure 2—figure supplement 1C*). All of the above are features hinted at lncRNA functional conservation across different species (*Quinn et al., 2016*).

Mouse lnc-*Nr2f1* is induced as early as 48 hr after BAM factors are expressed during MEF-to-iN cell conversion and peaks during mid-to-late stages of reprogramming (*Figure 2B*, *Figure 1—figure*

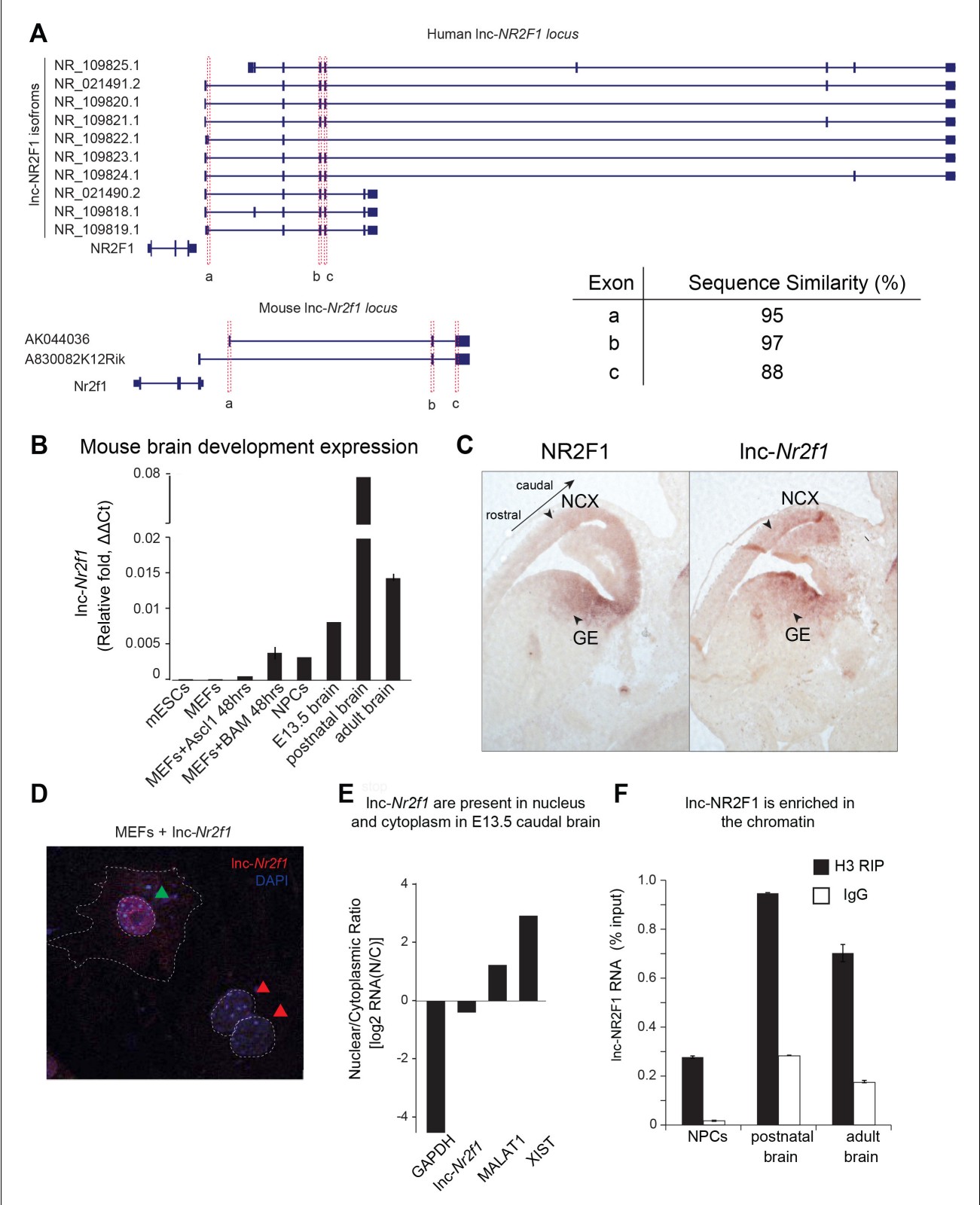

**Figure 2.** Molecular characterization of mouse *lnc-Nr2f1*. (A) Schematic showing the different isoforms reported by Refseq of the human lnc-*NR2F1* and mouse lnc-*Nr2f1*. Exons highlighted in red are conserved among human and mouse. The table at the bottom right corner shows the sequence similarity as reported by T-COFFEE. The sequence alignment is available as **Supplementary file 6**. (B) *Lnc-Nr2f1* expression measured by qRT-PCR across stages of mouse brain development and early stages of iN cell reprogramming. Results show early detection in E13.5 brain, peak expression at postnatal

*Figure 2 continued on next page*

*Figure 2 continued*

stages, and continued expression through adulthood. (C) In situ hybridization for *Nr2f1* and *lnc-Nr2f1* shows similar expression pattern in E13.5 mouse brain. Highlighted by arrows are neocortex (NCX) and ganglionic eminences (GE) with high expression levels. (D) Cellular localization of *lnc-Nr2f1* by single molecule RNA-FISH in MEFs ectopically expressing the lncRNA 48 hr after dox induction reveals nuclear and cytoplasmic localization, with slight nuclear enrichment. Green arrow points at lnc-*Nr2f1* in the nucleus and red arrows point at the uninfected nuclei. (E) Cellular fractionation of primary neurons derived from E13.5 caudal cortex dissection shows nuclear and cytoplasmic localization of *lnc-Nr2f1*. (F) Chromatin enrichment of *lnc-Nr2f1* by histone H3 RIP-qRT-PCR in brain derived neuronal precursor cells (NPCs), postnatal and adult mouse brain.

DOI: https://doi.org/10.7554/eLife.41770.007

The following figure supplements are available for figure 2:

**Figure supplement 1.** Synteny, sequence and microdomain conservation of lnc-*NR2F1*.

DOI: https://doi.org/10.7554/eLife.41770.008

**Figure supplement 2.** Characterization of *lnc-Nr2f1* localization.

DOI: https://doi.org/10.7554/eLife.41770.009

*supplement 2* and *Figure 2—figure supplement 2A*). In the developing and adult mouse brain, *lnc-Nr2f1* showed a distinct region-specific pattern of expression (*Figure 2C* and *Figure 2—figure supplement 2B*). In the developing telencephalon at E13.5, in situ hybridization with a probe against *lnc-Nr2f1* revealed strong expression in the caudolateral part of the mouse cortex and ganglionic eminences (GE), similar to *Nr2f1* expression (*Jonk et al., 1994*) (*Figure 2C* and *Figure 2—figure supplement 2B*).

To determine *lnc-Nr2f1*'s cellular localization, we performed single molecule RNA-FISH in MEFs ectopically expressing *lnc-Nr2f1*, which revealed a nuclear and cytoplasmic but predominantly nuclear localization (*Figure 2D* and *Figure 2—figure supplement 2C*). Consistently, cellular fractionation from primary neurons dissected from caudal region of the cortex showed endogenous localization of *lnc-Nr2f1* in both nuclear and cytoplasmic fractions (*Figure 2E*, *Figure 3—figure supplement 1H–I*). Within the nuclear fraction, *lnc-Nr2f1* is enriched in chromatin as assayed by histone H3 RNA Immunoprecipitation followed by qRT-PCR (histone H3 RIP-qRT-PCR) in brain-derived NPCs, postnatal and adult mouse brain (*Figure 2F*).

We next wanted to explore potential functional roles of *lnc-Nr2f1* and assessed its role during neuronal induction (*Figure 3—figure supplement 1*). We therefore co-expressed *lnc-Nr2f1* (NR_045195.1 or A830082K12Rik) with Ascl1 and asked whether it could promote neuronal conversion over Ascl1 alone as previously observed with other transcription factors (Brn2, Myt1l) (*Chanda et al., 2014*; *Treutlein et al., 2016*). To that end we infected MEFs with Ascl1 with or without *lnc-Nr2f1*, and determined the ratio of TauEGFP-positive cells with neuronal processes over the total number of TauEGFP-positive cells at day 7. We chose day seven to perform the experiment as it is an early time point for reprogramming. Indeed, the addition of *lnc-Nr2f1* showed an approximately 50% (1.5-fold) significant increase in the number of TauEGFP positive cells with neurites relative to Ascl1 alone (*Figure 3—figure supplement 1A–B*). This surprising morphological maturation phenotype were only previously observed only with co-expression of transcription factors (Brn2 and Myt1l) with Ascl1 demonstrating a role of *lnc-Nr2f1* in neuronal morphological maturation (*Mall et al., 2017*).

RNA-seq in sorted 7d MEF-iN cells expressing Ascl1, with and without co-expression of *lnc-Nr2f1*, revealed 343 genes significantly changed expression between data sets (RPKM >1, FDR corrected p<0.001) (*Figure 3—figure supplement 1C and E*). The vast majority of these genes were induced in expression upon *lnc-Nr2f1* expression, with 311 genes up- and 32 down-regulated, suggesting *lnc-Nr2f1* may positively enhance transcription. Gene ontology (GO) term enrichment of up regulated genes showed significant enrichment in biological functions related to plasma membrane (*extracellular region*, *cell adhesion*, and *transmembrane receptor tyrosine kinase activity*) and neuronal function (*neuron projection*, *calcium binding*, *neuron differentiation*, and *axonogenesis*) (*Figure 3—figure supplement 1D*). These pathways are consistent with phenotype observed for *lnc-Nr2f1* during iN cell reprogramming of promoting precocious maturation programs (*Figure 3—figure supplement 1D*). Amongst the up-regulated genes are well-characterized neuronal and axon guidance genes such as *NeuroD1*, *Gap43*, *Tubb4a*, *Ntf3*, *Nlgn3*, *Efnb3*, *Ntrk3*, *Bmp4*, *Sema3d*, *Slc35d3*, *Ror1*, *Ror2*, *Fgf7*. Additionally, genes previously associated with neurological disorders

were similarly up regulated, such as *Mdga2, Clu, Epha3, Chl1, Cntn4, Cdh23*, and *Pard3b* (*Basu et al., 2009*).

## *Lnc-Nr2f1* is required for proper neuronal gene expression

To investigate the contribution of *lnc-Nr2f1* to overall gene regulation, we sought to achieve *lnc-Nr2f1* gain and loss-of-function in one experimental system. We reasoned that mouse ES cells were the best way to accomplish loss-of-function. Since the functional domains of *lnc-Nr2f1* RNA are unknown, non-coding sequences cannot be turned into missense information by frame-shift mutations. Also, a large deletion encompassing the entire 20 kb *lnc-Nr2f1* locus may also inactivate interweaved intronic regulatory elements and chromatin structure. Instead, we chose to insert a polyA transcriptional termination signal to eliminate *lnc-Nr2f1* transcripts. We obtained mouse ES cells that were previously genetically characterized in a genome-saturating haploid ES cell mutagenesis screen (*Elling et al., 2017*). One of those ES cell clones had the mutagenesis cassette containing an inverted (therefore inactive) splicing acceptor and polyA site inserted after the first exon of *lnc-Nr2f1* ('Control' thereafter, *Figure 3A*). The mutagenesis cassette was designed to be conditionally reversible as it is flanked by combinations of loxP sites.

To achieve gain-of-function in the same cell system, we first overexpressed *lnc-Nr2f1* in mES cells together with the proneural Ngn2 because that was shown to efficiently and rapidly induce neurons from ES cells (*Zhang et al., 2013*). Again, we quantified the neurite length and neuron count, to test for its effect on increasing maturation kinetics. Consistent with the fibroblast reprogramming results, we saw a significant increase in neurite length upon *lnc-Nr2f1* overexpression though the number of neurons remained the same (*Figure 3B–D*). In contrast, overexpression of the coding gene *Nr2f1* did not induce these phenotypes, and instead caused a drastic reduction in the number of neurons (*Figure 3B–D*). These divergent results suggest that *lnc-Nr2f1* functions independently of *Nr2f1*. RNA-seq analysis showed lnc-Nr2f1 overexpression led to the induction of genes with functions in axon guidance (*Sema5d, Epha1)* and neuronal projection development (*Tubb6, Stmn2, Dtnbp1*) (*Figure 3E*), confirming the cell biology phenotype (FDR corrected p<0.10, fold change >1.5).

Next we turned to inactivate *lnc-Nr2f1* function in mouse ES cells. To generate an isogenic knockout line, we treated the conditionally mutant ES cell line with Cre recombinase, which resulted in an inversion of the polyA cassette, which in turn terminates *lnc-Nr2f1* transcription ('*lnc-Nr2f1* KO' thereafter) (*Figure 3F*). Since *lnc-Nr2f1* is not expressed in ES cells, we differentiated *lnc-Nr2f1* KO and control mES cells into induced neuronal cells by Ngn2 overexpression as above to assess the transcriptional consequences of loss of *lnc-Nr2f1* in a neuronal context. RNA-seq showed that 348 genes were differentially expressed between the control and the *lnc-Nr2f1* KO neurons, which can be subsequently rescued with *lnc-Nr2f1* overexpression (FDR corrected p<0.10) (*Figure 3G* and *Figure 3—figure supplement 1F*). Consistent with target genes in our *lnc-Nr2f1* overexpression study, we found *lnc-Nr2f1* KO led to down regulation of neuronal pathfinding and axon guidance genes such as *Sema6d* and proneural bHLH transcription factor *Neurod2* as well as deregulation of genes associated with autism spectrum disorder such as *Bdnf, Dcx* and *Nlgn3* (*Basu et al., 2009*) (*Figure 3G*). The transcriptional abnormalities were reversed by enforced expression of *lnc-Nr2f1* from a heterologous construct via lentiviral transduction. The rescue data indicate that the downregulation of neuronal genes and the upregulation of ectopic genes are caused by the loss of *lnc-Nr2f1* expression in the knock out cells and unlikely by disruption of the nearby DNA regulatory elements due to the insertion of the targeting cassette. Gene Ontology analysis of the downregulated genes in *lnc-Nr2f1* KO neurons revealed enrichment for terms related to neural functions (*regionalization, central nervous system development and neural precursor cell proliferation*), whereas the upregulated genes are enriched in development of non-neuronal tissues such as *circulatory system and skin development* (*Figure 3H and I*). Finally, the rescued genes overlap significantly with curated autism risk genes by Basu et. al. (*Basu et al., 2009*) (p=0.0012, Chi-square) (*Figure 3—figure supplement 1G*).

To further distinguish the function of *lnc-Nr2f1* vs *Nr2f1*, we generated *Nr2f1* heterozygous and homozygous KO mouse ES cell lines using CRISPR/Cas9 (*Figure 3—figure supplement 2A*). When we performed qRT-PCR on the day 4 iN cells generated from the Ctrl, *Nr2f1* heterozygous and homozygous null mES cells, we found that the level of both *Nr2f1* and *lnc-Nr2f1* RNA transcripts did not change (*Figure 3—figure supplement 2B*), However, protein quantitation confirmed that Nr2f1 protein level was reduced or eliminated in the heterozygous lines (clone 21 and 44) or homozygous

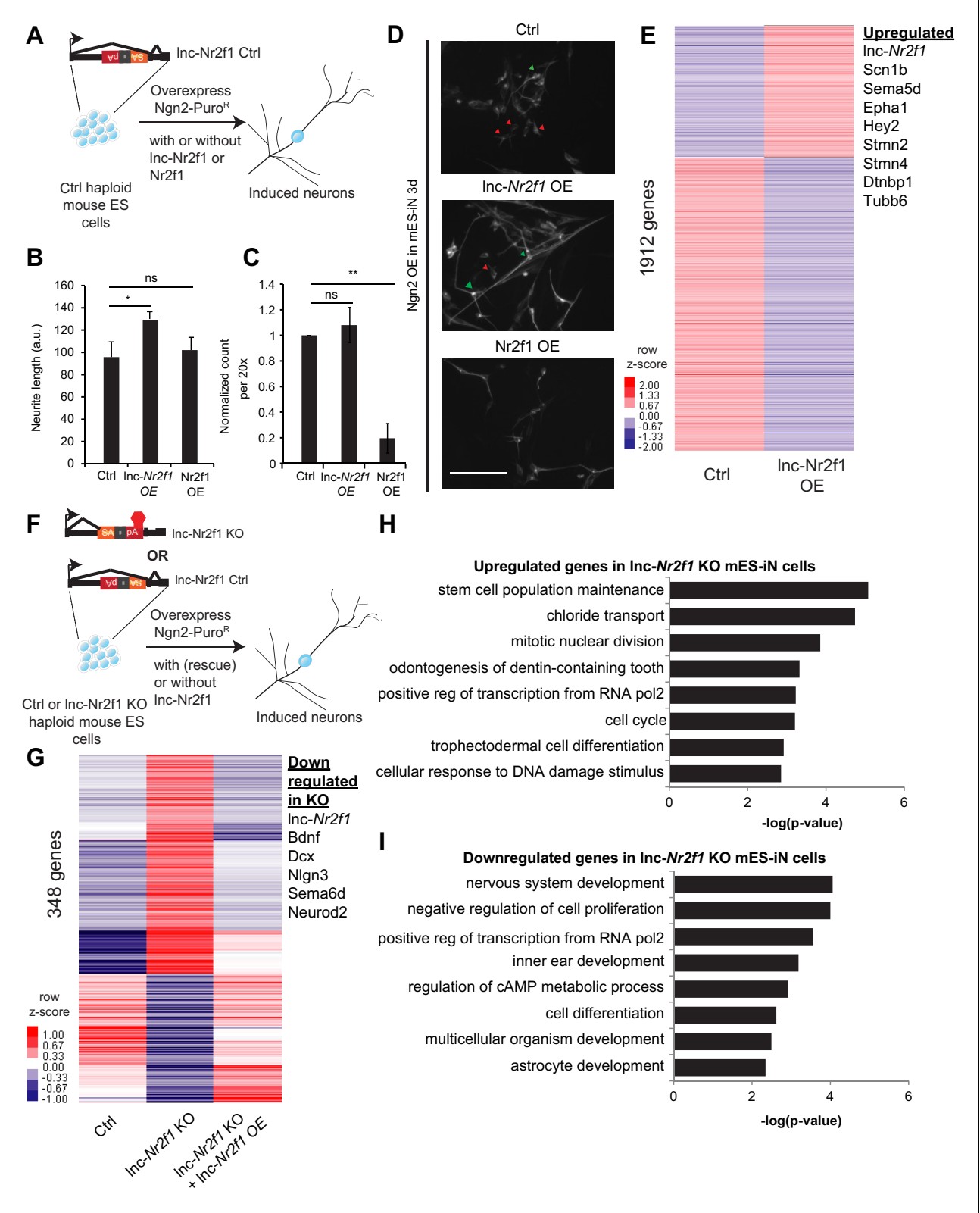

**Figure 3.** Mouse *Lnc-Nr2f1KO* reveals lnc-*Nr2f1* regulates neuronal genes. (**A**) Schematic showing the experimental strategy for lnc-*Nr2f1* overexpression. In control mouse ES cells, an inverted construct with a splice acceptor (marked in yellow) and a polyadenylation signal (marked in red) are added after the first exon of the *lnc-Nr2f1*. The mouse ES were infected with rtTA and Ngn2-T2A-puro and mES derived induced neurons (mES-iN) were assayed after 3 or 4 days after dox induction. (**B**) Graph showing that overexpression of *lnc-Nr2f1* increased the neurite length in day 3 Ngn2

*Figure 3 continued on next page*

*Figure 3 continued*

mouse ES derived iN cells relative to the Ctrl. The same effect was not seen with *Nr2f1* overexpression. For each replicate, the individual neurite length for all neurons in each of the five 20x field was manually traced in Fiji. The sequence used for mouse *lnc-Nr2f1* overexpression is available in the supplementary document (n = 3, Student t-test, Two-tailed, * indicates p=0.048 < 0.05). Error bars show s.e.m. (C) Graph showing that overexpression of *Nr2f1* decreased the neurite number in day 3 Ngn2 mouse ES-iN cells relative to the Ctrl. The same effect was not seen with *lnc-Nr2f1* overexpression. (n = 3, 10 field per replicate, Student t-test, Two-tailed, ** indicates p=0.0022 < 0.01). Error bars show s.e.m. (D) β-III-tubulin staining of the day 3 Ngn2 mouse ES derived iN cells for Ctrl, *lnc-Nr2f1* overexpression and Nr2f1 overexpression. Scale bar = 50 μm. Red arrow pointed at immature induced neuronal cells with short projection. Green arrow pointed at mature induced neuronal cells with longer projection. Note that the lnc-*Nr2f1* overexpression condition have more mature induced neuronal cells. (E) Hierarchical clustering heatmap of day 4 Ngn2 ES-iN cells between control and *lnc-Nr2f1* overexpression (OE). There are 1912 genes differentially expressed (n = 2, FDR corrected p<0.10, Fold change >1.5 fold). Listed to the right are genes which are upregulated upon *lnc-Nr2f1* overexpression. (F) Schematic showing the knocking out strategy for *lnc-Nr2f1*. The *lnc-Nr2f1* knockout mouse ES cells are generated after Cre recombinase introduction to the Ctrl line in **Figure 3A**. The mouse ES were infected with rtTA and Ngn2-T2A-puro and mES derived induced neurons were assayed after 3 or 4 days after dox induction. (G) Hierarchical clustering heatmap of day 4 Ngn2 ES-iN cells between wild type, *lnc-Nr2f1* knockout (KO) and *lnc-Nr2f1* knockout with *lnc-Nr2f1* overexpression (OE). There are 348 genes differentially expressed and can be subsequently rescued with *lnc-Nr2f1* overexpression (n = 2, FDR corrected p<0.10). Listed to the right are genes which are upregulated upon *lnc-Nr2f1* KO. (H) Gene ontology of the upregulated genes in *lnc-Nr2f1* knockout day 4 Ngn2 mouse ES- iN cells as compared to the Ctrl. (I) Gene ontology of the downregulated genes in *lnc-Nr2f1* knockout day 4 Ngn2 mouse ES- iN cells as compared to the Ctrl.

DOI: https://doi.org/10.7554/eLife.41770.010

The following figure supplements are available for figure 3:

**Figure supplement 1.** Characterization of the roles of *lnc-Nr2f1* during iN reprogramming.

DOI: https://doi.org/10.7554/eLife.41770.011

**Figure supplement 2.** Characterization of the epistasis relationship between mouse *Nr2f1* and *lnc-Nr2f1*.

DOI: https://doi.org/10.7554/eLife.41770.012

null clones (Clone 2, 11 and 18), respectively (**Figure 3—figure supplement 2C**). The *Nr2f1* KO did not affect *lnc-Nr2f1* expression (**Figure 3—figure supplement 2B**) and had no impact on the neurite length or number in mES-iN cells (**Figure 3—figure supplement 2D–E**). In summary, both gain and loss of function studies demonstrated that *lnc-Nr2f1* plays a role in the transcriptional regulation of a gene network involved in neuronal maturation pathways that ultimately resulted in faster acquisition of a mature neuronal identity in both MEFs and mES cells and is functionally distinct from its neighboring coding gene, *Nr2f1*.

## Mouse *lnc-Nr2f1* binds to distinct genomic loci regulating neuronal genes

As described above, histone pull-down experiments suggested an association of lncNr2f1 with chromatin. We therefore sought to map the precise lnc-*Nr2f1* genome wide occupancy and performed Chromatin Isolation by RNA precipitation followed by sequencing (ChIRP-seq) on day 4 mES-iN (**Figure 4A**, **Figure 4—figure supplement 1A–B**). To minimize background sequencing we used even and odd probes targeting *lnc-Nr2f1* in replicate experiments and only considered the overlapping peaks from both experiments (**Figure 4B**). Both even and odd probes pulled down *lnc-Nr2f1* efficiently. There are 14975 peaks called by MACS, and the peak signals are consistent between replicates (n = 4) with little signal in RNase-treated control samples (**Figure 4B**). We obtained 1092 high confidence peaks with further filtering for the most significant and reproducible binding events (see Materials and methods for filtering criteria). As an example, *lnc-Nr2f1* binds to the intronic region of *Nrp2*, a gene with known roles in neuronal pathfinding (**Figure 4C**). To understand which gene ontology terms are enriched in the genes adjacent to the mES-iN ChIRP peaks, we performed GREAT analysis and associated the 1092 *lnc-Nr2f1* binding sites to 1534 genes. GO term analysis revealed that these genes that enriched for neuronal terms such as *central nervous system development, synapse organization and chemical synaptic transmission* (**Figure 4D**). Using the publicly available ChIP-seq peak sets for CTCF, enhancer, H3K27ac, H3K4me3 and PolII obtained from mouse adult cortex and E14.5 brain, we found significant enrichments of those peaks co-localizing with the enhancers, H3K27ac and H3K4me3 marks relatively to the background (**Figure 4E** and **Figure 4—figure supplement 1D**) (**Shen et al., 2012**). DNA motif analysis of *lnc-Nr2f1* binding sites revealed several basic helix loop helix (bHLH) factor motifs that are significantly enriched (*NeuroD1, Atoh1, Olig2 and Ptf1a*) (**Figure 4—figure supplement 1C**). To test whether lnc-Nr2f1 and the bHLH factors are binding to same genomic regions, we compared ChIP-seq data of bHLH neurogenic factors

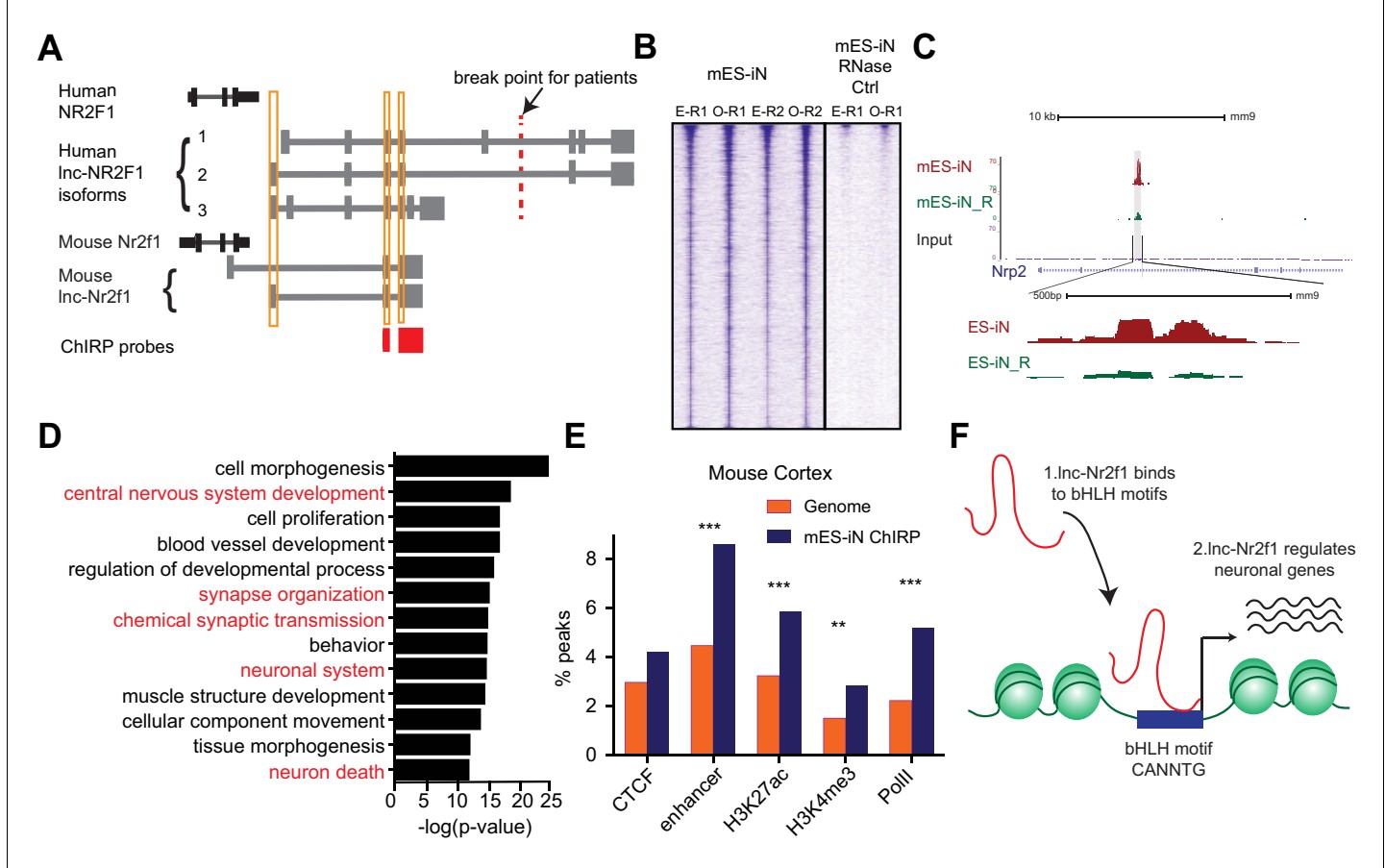

**Figure 4.** *lnc-Nr2f1* binds to distinct genomic loci regulating neuronal genes. (**A**) Schematic showing the location of ChIRP probe for mouse *lnc-Nr2f1* (highlighted in red). Yellow lines represent the conserved exons between mouse and human lnc-*Nr2f1*. (**B**) Heatmaps representing genome-wide occupancy profile for mouse *lnc-Nr2f1* in day 4 Ngn2 mouse ES- iN cells and the RNase control obtained by ChIRP. There are 14975 significant peaks called with respect to the RNase treated control. E and O represents even and odd probes respectively. (**C**) UCSC browser track showing the binding site within the intronic region of Nrp2. The 'R' represents the RNase treated control. (**D**) Gene ontology terms associated with genes adjacent to the high confident mES-iN ChIRP-seq peaks. Terms highlighted in red are terms related to nervous system development. (**E**) Percentage of mES-iN ChIRP-seq peaks which overlap with CTCF, enhancer, H3K27ac, H3K4Me3 and PolII defined in mouse cortex relative to the control. (*** represents p<0.0001, ** represents p<0.01, Chi-square test) (**F**) Proposal mechanism of lnc-*Nr2f1* action. lnc-*Nr2f1* binds to the genomic region enriched with bHLH motif and regulates the downstream neuronal genes.

DOI: https://doi.org/10.7554/eLife.41770.013

The following figure supplement is available for figure 4:

**Figure supplement 1.** Identification of *lnc-NR2F1* role in transcriptional regulation.
DOI: https://doi.org/10.7554/eLife.41770.014

(Ngn2 and Ascl1) with lnc-Nr2f1 ChIRP-seq data (*Figure 4—figure supplement 1F,G*). We found that direct overlap between lncNr2f1 ChIRP-seq peaks and ChIP-seq peaks are low but statistically significant. 3.2% and 13.2% lncNr2f1 ChIRP-seq peaks overlapped with Ngn2 and Ascl1 ChIP peaks, respectively (*Figure 4—figure supplement 1F*). For peak-associated genes, lncNr2f1 target genes overlap very significantly with Ngn2 (37.9%) or Ascl1 (67.3%) target genes (*Figure 4—figure supplement 1G*). These results suggest that lncNr2f1 and bHLH transcription factors such as Ascl1 and Ngn2 may coordinately regulate the same set of neuronal genes, and the majority of instances occur with lnc-Nr2f1 and the bHLH factors binding nearby but non-overlapping sites. Finally, to understand whether the *lnc-Nr2f1* regulates the genes listed in *Figure 3E*, we overlapped the 1534 genes adjacent to the mouse *lnc-Nr2f1* ChIRP peaks with the genes up or downregulated upon *lnc-Nr2f1* over-expression, we found 177 common genes between the two lists (p<0.0001, Chi Square), suggesting

that those genes might be direct lnc-Nr2f1 targets since they are occupied by lnc-Nr2f1 RNA and are significantly altered when manipulating lnc-Nr2f1 (*Figure 4—figure supplement 1E*)

## Human *lnc-NR2F1* shows isoform-specific chromatin binding

The balanced chromosomal translocation t(5;12) detected in patient CMS12200 disrupts the long *lnc-NR2F1* while the short isoform appears unaffected. We therefore hypothesized that the *lnc-NR2F1* might have isoform-specific functions and the long isoforms are contributing to the phenotype observed in patient CMS12200. Given that *lnc-NR2F1* is highly expressed in human brain tissue and it has high sequence conservation between mouse and human (*Figure 5A*), we next sought to determine whether human *lnc-NR2F1* had a similar role in neuronal reprogramming as the mouse transcript and whether the different isoforms may have distinct functions. Therefore, we individually expressed each of the three human *lnc-NR2F1* isoforms in MEFs, and measured their ability to enhance Ascl1- mediated neuronal reprogramming, as judged by morphological complexity of TauEGFP cells (*Figure 5B*). The long human *lnc-NR2F1* isoform two significantly increased the proportion of TauEGFP cells with projections, albeit with a slight smaller magnitude than the mouse lncRNA. Intriguingly, long isoform one inhibited neuronal maturation while the short isoform three had no significant effect (*Figure 5B*). Thus, different isoforms of *lnc-NR2F1* may possess differential regulatory activity. Long *lnc-NR2F1* isoforms disrupted by chromosomal translocation in patient CMS12200 can impact neuronal maturation, while the short *lnc-NR2F1* isoform remaining intact in patient CMS12200 did not have a detectable effect on neuronal complexity.

Due to *lnc-NR2F1*'s strong association with chromatin and isoform-specific function, we hypothesized that different domains of *lnc-NR2F1* may have differential chromatin localization. To test this idea, we mapped the genome-wide localization of different RNA domains in *lnc-NR2F1* using domain-specific Chromatin Isolation by RNA precipitation followed by sequencing (domain ChIRP-seq) (*Shen et al., 2012*; *McLean et al., 2010*). We performed *lnc-NR2F1* ChIRP-seq in human neural progenitor cell (hNPC) differentiated 12d from human embryonic stem cells using dual SMAD inhibition protocol (*Chambers et al., 2009*) (*Figure 5—figure supplement 1A*). We performed ChIRP-seq separately with two orthogonal probe sets (termed odd and even sets) against two different domains of *lnc-NR2F1* (long isoform-specific exon 11 and the short isoform-specific exon 7) and only accepted concordant results between the odd and even probe sets. There are approximately 10-fold more genomic occupancy for the long vs. short isoform of *lnc-NR2F1*: 4404 ChIRP-seq peaks for exon 11 (n = 4) vs. 415 ChIRP-seq peaks for exon 7 (n = 4), respectively. It is unlikely that low expression or inefficient pulldown of the short isoform are the cause of the difference given that we detected comparable level of long and short isoform in hNPC and obtained similar recovery of the long and short RNA isoform (*Figure 5—figure supplement 1B and C*). Consistent with our hypothesis of domain specific chromatin localization, genomic occupancy sites of different *lnc-NR2F1* exons showed limited overlap of peaks (*Figure 5C*), suggesting that the long- and short- specific exon might be each binding to different genomic loci and regulating different subsets of downstream genes. For example, only the long isoform-specific exon has a distinct binding site surrounding *POLR1A* (*Figure 5D*).

We then assessed the transcriptional response which the three isoforms of human *lnc-NR2F1* varied quantitatively: The shortest isoform, human *lnc-NR2F1* isoform 3, had the lowest number of differentially regulated genes (5 downregulated and four upregulated) compared to isoform 1 (1147 downregulated and 414 upregulated) and isoform 2 (141 downregulated and 45 upregulated) (*Figure 5—figure supplement 1F*). This is consistent with our hypothesis that the long isoform is the functional one in neurogenesis and in patients with neurodevelopmental disorders including ASD.

After further filtering the peaks for high confidence peaks, we obtained 913 high confidence peaks for long-isoform domain ChIRP and no peaks for short-isoform domain ChIRP (see Materials and methods for filtering criteria). To further characterize *lnc-NR2F1* occupancy patterns, high confidence ChIRP peaks were classified according to distance to putative *cis*-genes (*Figure 5E* and *Figure 5—figure supplement 1D*). Relative to the human genome, the long isoform-specific peaks are more enriched in the exonic, intronic, enhancer and promoter regions and depleted in the intergenic regions (*Figure 5E*). Furthermore, ChIRP peaks are characterized by chromatin state model, which defines human genome with 25 chromatin states using 12 biochemical features (histone modifications, DNA accessibility, DNA methylation, RNA-seq and other epigenetic signals) (*Matsui et al., 2012*). *Lnc-NR2F1* ChIRP peaks are enriched in promoter, enhancer, and transcribed

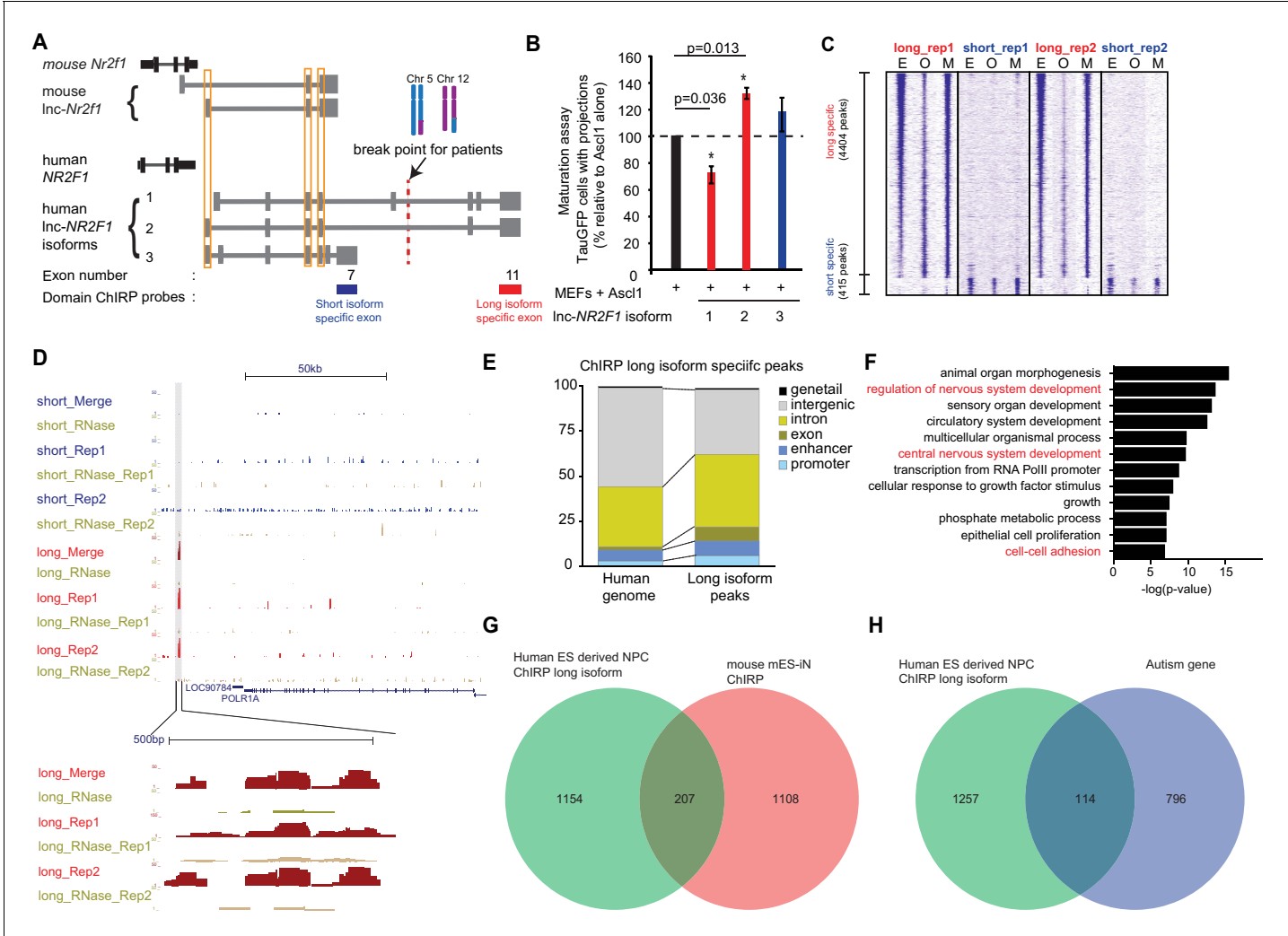

**Figure 5.** Human *lnc-NR2F1* shows isoform-specific chromatin binding. (**A**) Schematic showing the location of the ChIRP probes target the short isoform-specific exon (exon 7) and long isoform-specific exon (exon 11). The red line denotes the break point for the patient. (**B**) Overexpression of human *lnc-NR2F1* isoforms in combination with Ascl1 relative to MEFs expressing Ascl1 alone. The graph quantifies the proportion of TauGFP positive cells with projections normalized to number of TauGFP cells. TauGFP cells with projections longer than three times the diameter of the cell body were counted and normalized to the total number of TauGFP positive cells. The sequences for human *lnc-Nr2f1* isoforms is available in the supplementary documents (n = 3, Student t-test, two tailed, scale bar= * represents p<0.05, ** represents p<0.01). Error bars show s.e.m. (**C**) Heatmap representing genome-wide occupancy profile for domain ChIRP performed using probes specific to the long and short isoform-specific exon of *lnc-NR2F1* in human ES derived neural progenitor cells (NPC). There are 4404 and 415 significant peaks called relative to the RNase control for the long and short isoform respectively. E, O and M represents even, odd and merge track respectively. (**D**) UCSC browser track showing the site within the promoter region of LOC90784 bound by the long isoform-specific exon (exon 11) but not the short isoform-specific exon (exon 7) (**E**) Bar graph showing the distribution of the 913 high confident long isoform-specific peaks. The long isoform-specific peaks are enriched in the introns, exons, promoters and enhancers but depleted in the intergenic regions. (**F**) Gene ontology terms associated with genes adjacent to the human ES derived NPC ChIRP-seq high confident peaks. Terms highlighted in red are terms related to nervous system development. (**G**) Venn diagram representing the peak associated gene overlap between the domain ChIRP of the long isoform-specific exon (exon 11) from human ES derived NPC and mouse mES-iN ChIRP. (p<0.0001, χ2 = 239.921, DF = 1, Chi square test) (**H**) Venn diagram representing overlap between genes involved in the autism risk and genes identified by the domain ChIRP of the long isoform-specific exon (exon 11) from human ES derived NPC. (p<0.0001, χ2 = 71.670, DF = 1, Chi square test).
DOI: https://doi.org/10.7554/eLife.41770.015

The following figure supplement is available for figure 5:

**Figure supplement 1.** Identification of *lnc-NR2F1* role in transcriptional regulation.
DOI: https://doi.org/10.7554/eLife.41770.016

regions, compared to the whole genome which majority is in the quiescent state (*Figure 5—figure supplement 1D*).

The 1361 high confidence peak associated genes by the long isoform-specific domain ChIRP are all enriched for neuronal specific biological terms such *central nervous system development, cell-cell adhesion* and *regulation of nervous system development* (*Figure 5F*). There is also a significant overlap between the genes adjacent to peaks from the long isoform-specific domain ChIRP and the genes adjacent to the mouse *lnc-Nr2f1* ChIRP, indicating a possible conserved functions of *lnc-Nr2f1* between mouse and human (207 genes overlapped, Chi-square test, p<0.0001) (*Figure 5G*). We also observed a significant overlap between the genes adjacent to the peaks from the long isoform-specific domain ChIRP and the autism risk genes (114 genes, Chi-square test, p<0.0001) (*Figure 5H*). Notably, peaks of long isoform-specific exon 11 ChIRP are enriched for multiple basic-helix-loop-helix motifs (Ascl1, NeuroD1, Olig2 and Atoh1) which all share the CANNTG motif (*Kim et al., 2011*) (*Figure 5—figure supplement 1E*). Mouse ES-iN cell ChIRP peaks are also enriched for similar motifs suggesting possibly conserved mechanisms of *lnc-NR2F1* in human and mouse. Given the pervasive roles of bHLH proteins in neuronal development and in induced neuronal reprogramming, the binding preference of *lnc-NR2F1* suggests a biochemical basis for the functional cooperativity with proneural bHLH factors (Ngn2 and Ascl1). In summary, we conclude that the *lnc-NR2F1* isoforms have different genomic occupancy and transcriptional effects. The long isoform showed most biological activity and chromatin binding and is also solely affected in patient CMS12200.

## Discussion

Given the stringency required to rewire a cellular state from an unrelated lineage, factors expressed during direct reprogramming likely have an active role in the establishment of the new cell identity. Direct neuronal lineage induction represents a synchronized and streamlined conversion of cell fate, and should be a powerful system to enrich for lineage-specific regulatory factors. In this study, direct conversion of fibroblasts to induced neuronal cells enabled the identification of lncRNAs with unique properties in establishing neuronal fate via neurogenesis or maturation programs. The pipeline described in this study may be extrapolated to identify potential lncRNA regulators of other specific cellular states.

Because the brain is the organ with the greatest number of unique cell-type specific lncRNAs (*Qureshi et al., 2010*), our approach may be useful to identify lncRNAs with roles in neural lineage specification. Indeed, we identified *lnc-Nr2f1* as functional player in neuronal maturation and pathfinding. Remarkably its sequence is remarkably conserved between the first few exons of mouse and human *lnc-NR2F1* which is an atypical pattern for lncRNAs (*Ulitsky and Bartel, 2013*; *Ulitsky et al., 2011*). Consequently, we found that there is high synteny, sequence and microdomain conservation between mouse and human *lnc-NR2F1*. These observations suggest that some lncRNAs may have been functionally conserved throughout evolution.

In this study, we focus on the functional characterization of *lnc-NR2F1* locus because it is recurrently mutated in human patients with ASD/ID. We identified a patient, whose genome harbors a balanced translocation disrupting the *lnc-NR2F1* locus without any other detectable pathogenic genetic variations and shows abnormal neurodevelopmental symptoms; therefore, implicating this lncRNA as a critical regulator of brain development and function. The father of the proband carries the same translocation and suffers from dyslexia and stuttering, suggesting that the phenotype may be transmitted in a Mendelian manner. However, the much milder phenotype of the father implies that additional genes, environmental factors, or compensatory neuronal circuitry acquired during adulthood may influence the severity of the outcome.

Given the close genomic proximity of *lnc-NR2F1* and *NR2F1* increased attention must be devoted to consider the possibility of a contribution of the coding gene. Since unknown regulatory elements for the coding gene could be affected the human genetics data are not decisive. However, several functional experiments point to a contribution of *lnc-NR2F1* rather than the coding gene. First, gain and loss of function studies as well as chromatin localization clearly show that *lnc-NR2F1* acts in trans to affect gene expression. Second, we can rescue the phenotype of the *lnc-Nr2f1* KO by overexpression of *lnc-Nr2f1* mRNA. Third, *lnc-Nr2f1* overexpression in *lnc-Nr2f1* KO cells does not affect *Nr2f1* expression. Fourth, ChIRP-sequencing of *lnc-Nr2f1* in induced neurons derived from mES cells does

not show binding of *lnc-Nr2f1* in the *Nr2f1* promoter region (**Figure 4F**). The only definitive answer may be obtained from human postmortem tissue analysis of affected patients.

Neurodevelopmental and neuropsychiatric disorders are complex diseases manifesting in a spectrum of phenotypes. We integrated lncRNA expression pattern, in vitro functional screen, and human genetic data to pinpoint potentially causal lncRNAs. We concentrated on genomic lesions affecting lncRNAs, which have been largely understudied regulatory factors in these diseases, and connected them to specific phenotypes. We found several of the lncRNA candidates were disrupted by focal chromosomal aberrations in patients diagnosed with ASD/ID, establishing a link between human disease and lncRNA function. The advent of next generation sequencing has greatly improved the ability to pinpoint causal disease mutations in protein coding genes including the discovery of novel autism genes. Most of the other functional regions of the genome, however, have largely been ignored as part of exome sequencing approaches. While full genome sequencing of patients is beginning, functional interpretation remains a daunting challenge. We present a strategy to begin to characterize the functionally important non-coding regions as it relates to disease. Our work highlights lncRNA mutations as an understudied and important potential next frontier in human genetics related to neurodevelopmental disease.

## Materials and methods

**Key resources table**

| Reagent type (species) or resource | Designation | Source | Identifier | Additional information |
|---|---|---|---|---|
| Antibody | Rabbit polyclonal anti-H3 | Abcam | ab1791 (RRID:AB_302613) | |
| Antibody | Goat polyclonal anti-Sox1 | R and D | AF3369 (RRID:AB_2239879) | IHC (1:50) |
| Antibody | Rabbit polyclonal anti-β-III-tubulin | Covance | Discontinued | IHC (1:1000) |
| Antibody | Mouse monoclonal anti-Nestin | R and D | MAB1259 (RRID:AB_2251304) | IHC (1:1000) |
| Antibody | Rabbit monoclonal anti-HSP90 | Cell Signalling | 4877 (RRID:AB_2233307) | WB (1:2500) |
| Antibody | Rabbit monoclonal anti-Nr2f1 | Cell Signalling | 6364 (RRID:AB_11220432) | WB (1:1000) |
| Chemical compound, drug | SB431542 | Tocris | 1614 | |
| Chemical compound, drug | LDN198189 | MiliporeSigma | 5.09882.0001 | |
| Chemical compound, drug | CHIR99021 | StemGent | 04–0004 | |
| Chemical compound, drug | PD0325901 | Axon | 1408 | |
| Chemical compound, drug | Leukemia Inhibitory Factor | Generated in the lab | | |
| Cell line (H. Sapiens) | Human: 293T | ATCC | CRL-3216 (RRID:CVCL_0063) | |
| Cell line (H. Sapiens) | Human: H9 hESC line | UWisconsin | H9 (RRID:CVCL_9773) | |
| Cell line (H. Sapiens) | Human: SK-N-SH | ATCC | HTB-11 (RRID:CVCL_0531) | |
| Cell line (Mus musculus) | Tau: EGFP Mouse embryonic fibroblasts | Generated in the lab | | |
| Cell line (Mus musculus) | Mouse: Haploid ES cells | Obtained from Penninger lab (Elling et al. 2011) | | |

*Continued on next page*

*Continued*

| Reagent type (species) or resource | Designation | Source | Identifier | Additional information |
|---|---|---|---|---|
| Genetic reagent (*M. musculus*) | B6.129S4(Cg)-*Mapt*<sup>tm1(EGFP)Klt</sup>/J | Jackson | 29219 (RRID:IMSR_JAX:004779) | |
| Recombinant DNA reagent | TetO-*lnc-Nr2f1* (Mouse) | This paper | | |
| Recombinant DNA reagent | TetO-*lnc-NR2F1*-I PGK blast (Human) | This paper | | |
| Recombinant DNA reagent | TetO-*lnc-NR2F1*-II PGK blast (Mouse) | This paper | | |
| Recombinant DNA reagent | TetO-*lnc-NR2fF1*-III PGK blast (Mouse) | This paper | | |
| Recombinant DNA reagent | TetO-NR2F1 (Mouse) | This paper | | |

This method section was organized into four categories: animal and human protocols, cell culture, computational and sequencing methods and biochemistry. Within each category, method descriptions were arranged in the order they appear in figures. The ChIRP probes, public datasets, qRT-PCR primers and RNA FISH probes are available in *Supplementary files 2*, *3*, *4* and *5* respectively.

## Animal and human protocols
### Animal
All mouse work was performed according to IACUC approved protocols at Stanford University. Samples in the paper were obtained without determining their sex. All animals were housed in an animal facility with a 12 hr light/dark cycle.

### Human subjects
The study protocol, consent form, consent to publish and privacy practices were reviewed and approved by the Institutional Review Board of the Self Regional Healthcare, Greenwood, SC (Reference number Pro00074882).

## Cell culture and tissue dissection
### Cell culture
All cell lines (SK-N-SH, 293T) were purchased from ATCC and were verified by the manufacturer by STR profiling. They were also screened for mycoplasma and cultured using recommended conditions. Mouse embryonic fibroblasts (MEF) were derived from E13.5 Tau: :EGFP embryos and cultured in MEF media [500 ml of DMEM (Gibco), 50 ml of Cosmic Calf Serum (Thermo Scientific), 5 ml of Non-essential amino acid, 5 ml of Sodium Pyruvate, 5 ml of Penicillin/Streptomycin (Gibco), 4 ul of β-mercaptoethanol (Sigma)].

Mouse haploid embryonic stem cells were cultured in mouse embryonic stem cell media [341.5 ml DMEM (Gibco), 50 ml Knockout Serum Replacement (Gibco), 12.5 ml of Cosmic Calf Serum (ThermoScientific), 4.2 ml of Penicillin/Streptomycin, 4.2 ml of Non-essential amino acid, 4.2 ml of Sodium Pyruvate (Gibco), 4 ul of β-mercaptoethanol (Sigma) with leukemia inhibitory factor, 3 µM of CHIR99021 and 1 µM of PD3259010 (Both Tocris, Final concentration)].

Human embryonic stem cells (H9, University of Wisconsin) were cultured in mTESR media (Stem Cell Technologies). The experiments were performed in accordance with California State Regulations, CIRM Regulations and Stanford's Policy on Human Embryonic Stem Cell Research.

## Mouse postnatal/adult brain dissection

Briefly, forebrains were dissected from TauGFP heterozygous E13.5 embryos in cold HBSS, triturated in DMEM/F12 media, filtered through a 70 um filter and cultured in monolayer. Neural stem cells (NSC) were propagated in DMEM/F12 with N2 and B27 supplements (Invitrogen) with 20 ng/ml of FGF2 and 10 ng/ml of EGF. Postnatal brains (Postnatal day 0) and adult brains (three weeks old) were obtained from C57BL6 mice. To obtain postnatal brains, pups were anaesthetized in an ice bath before the whole brain was removed. To obtain adult brains, mice were euthanized using cervical dislocation before dissecting the whole brain out. For both adult and postnatal brains, they were manually dissociated to fine pieces before being digested in 0.25% trypsin for 30 min. They were triturated from time to time until a clear suspension was obtained. The cells were spun down at 1000 rpm for 5 min before proceeding to glutardehyde fixation.

## Reprogramming of mouse fibroblasts to induced neuronal cells (iN cells)

We followed protocols previously described (Wapinski et al., 2013). Briefly, mouse embryonic fibroblasts harvested from E13.5 Tau: :EGFP embryos were plated at a density of 25000 cells/cm$^2$. The next day, lentiviruses carrying TetO-FUW-ASCL1 and FUW-rtTA were added. Doxycycline (Final concentration: 2 µg/ml, Sigma) in MEF media was added to the wells. Media was changed to neuronal media [N2 +B27+DMEM/F12 (Invitrogen) +1.6 ml Insulin (6.25 mg/ml, Sigma)]+doxycycline two days after doxycycline induction. Subsequently, media was changed every three days. To obtain a pure population of day 7 TauEGFP positive neurons, the cells were digested using 0.25% trypsin (Invitrogen) and subjected to FACS. Forward and side scatters were used to exclude doublets and dead cells. The gating for GFP was set with a negative control (MEF).

## Maturation screen for *lnc-NR2F1*

To examine whether the lncRNA candidates can facilitate mouse embryonic fibroblasts (MEFs) to induced neuronal cells reprogramming, mouse and the three human isoforms for *lnc-NR2F1* were synthesized (sequences in the supplementary documents) and cloned into the TetO-FUW or TetO-PGK-blast$^R$ backbone respectively (available from Addgene). To examine whether those lncRNA candidates can help facilitate the maturation process, the number of MAP2 positive neuronal cells with projections three times the diameter of the cell body was counted at day seven and normalized to the total number of MAP2 positive cells. For neurite length measurement, Simple Neurite Tracer (ImageJ) was used manually to track neurite.

## Reprogramming of mouse embryonic stem cells to induced neurons

We followed the protocol previously described (*Zhang et al., 2013*). Mouse embryonic stem cells were plated single cell and infected the next day with TetO-NGN2-T2A-PURO$^R$ and FUW-rtTA. Doxycycline was added to the wells the next day. To select for only Ngn2 transducing cells, puromycin (Final concentration: 2 µg/ml, Sigma) was added in addition to doxycycline the next day and kept for 3 days.

## Generating *lnc-Nr2f1* KO ES-iN cells

We obtained mouse ES cells that were previously generated in a genome-saturating haploid ES cell mutagenesis screen (*Chu et al., 2011*). We identified one ES cell clone had the mutagenesis cassette containing a splicing acceptor and polyA site inserted after the first exon of *lnc-Nr2f1*. The orientation of the polyA site is in reverse from the transcription direction of *lnc-Nr2f1* so it's non-disruptive. The insertion is confirmed by PCR and sequencing. For generating *lnc-Nr2f1* KO clones, we did nucleofection of cre recombinase to invert polyA cassette since the polyA cassette is flanked by loxP sites. After nucleofection, we plated cells at low density and picked single colonies for testing the polyA inversion. The control and KO clones were then expanded for a few passages, allowing majority of them to become diploid cells. The homozygous diploid cells were then plated at 300 K cells/6 well in mES media at day 0. They were then infected with TetO-Ngn2-T2a-puro, FUW-rtTA and TetO-GFP the next day. At day 2, the media was changed to neuronal media [N2 +B27+DMEM/F12 (Invitrogen) +1.6 ml Insulin (6.25 mg/ml, Sigma)] and doxycycline (Sigma, Final conc: 2 µg/ml) was added. At day 3, puromycin (Sigma, 2 ug/ml) was added to the neuronal +dox media. RNA was harvested four days after dox induction.

## Generating *Nr2f1* KO ES-iN cells

For CRISPR/Cas9 genome editing of Nr2f1, gRNAs targeting second exon of Nr2f1 are cloned to a plasmid (pSpCas9(BB)−2A-Puro, pX459, Addgene #62988) expressing both the Cas9 protein and the gRNA. gRNA sequences were designed using the online tool (http://crispr.mit.edu/) provided by the Zhang lab (gRNA sequence used: CATGTCCGCGGACCGCGTCG). 24–48 hr after ES cell nucleofection, puromycin was added to select for 2–3 days. The remaining cells were plated at plate 100, 300, 1000, 3000 cells per plate for picking single colonies. Genomic DNA of each single colony was extracted using QuickExtract DNA Extraction Solution (Epicentre, QE09050). This extract was then used in a PCR of the genomic region that had been targeted for knock out (Fwd primer: AGAGA-CACCTGGTCCGTGAT. Reverse primer: GAGCCGGTGAAGGTAGATGA). PCR products were then Sanger sequenced to identify clones that would result in frameshifts and truncated Nr2f1. Sequence alignment and genomic PCR primer design was carried out using SnapGene software and cutting efficiency is calculated using web tool TIDE (https://tide-calculator.nki.nl/).

## Computational and sequencing methods

### LncRNA discovery pipelines (related to *Figure 1—figure supplement 1*)

TopHat was used for de novo alignment of paired-end reads for each of the samples. An assembled transcriptome was built from merged time points using Cuffmerge function. The de novo iN transcriptome was compared to RefSeq genes and annotated protein coding genes were removed, while non-coding genes annotated as 'NR' were kept. Expression level of genes was calculated in unit of fragments per kilobase of exon model per million mapped fragments (FPKM). Genes with low FPKM (average log2 FPKM across all samples less than 1) were removed. Genes with $p$-value<0.05 and at least two-fold expression change during iN reprogramming were defined as significant.

### Histone H3 RIP-seq (related to *Figure 1—figure supplement 1F*)

RNA isolated from H3 RIP was amplified and converted to cDNA using Nugen Ovation RNA-Seq System V2. The product was sheared using Covaris to 100–300 bp. Libraries were prepared using SPRIworks system for Illumina sequencing. The following antibodies were used for RIP: Rabbit anti-H3 (Abcam ab1791) and rabbit IgG (Abcam ab37415). For the H3 RIP-seq analysis we used a similar pipeline to bulk RNAseq assays. We first remove duplicate reads, clip adaptor sequences, discard short reads. Reads were then aligned to mm9 using Tophat2. Using samtools we convert the files from sam file to bam format. Filtered reads are normalized to sequence depth, and subsequently we calculate RPKM. The RIP-seq experiment was conducted with H3 and IgG antibodies. We sequenced both libraries and also 1% input material. To determine whether a lncRNA is enriched we compute the number of reads from H3 relative to input, and IgG relative to input using an in-house Perl script (rnaexp_rpkm.pl). We then calculated the fold-enrichment between H3 and IgG RIPs. Since the background was very low, anything greater than 2-fold and $p<0.05$ between H3 and IgG was considered enriched. Experiments in NPCs, MEFs, and adult brain were conducted in biological replicates. Only lncRNAs reproducibly enriched in the H3 RIP from biological replicates were considered chromatin associated and display as binary in the *Figure 1—figure supplement 1F*.

### Co-expression analysis for lncRNAs (related to *Figure 1—figure supplement 1G*)

We first obtained mouse RNAseq data from ENCODE, and calculated the RPKMs for all transcripts including coding and non-coding. We then for each non-coding RNA, calculated the Pearson correlation of the non-coding RNA with every coding transcript. If the correlation is greater than 0.5, this non-coding RNA was defined as positively correlated with the coding gene, and if the correlation is less than 0.5, it was defined as negatively correlated. We then obtained a matrix of coding genes versus non-coding genes, with positive (+1) and negative (−1) correlations as values in the matrix. We then use the GeneSets function in Genomica software from (http://genomica.weizmann.ac.il/), and generated a enrichment (-log(p-values)) matrix for iN lncRNAs associated with Gene Ontology terms based on similar expression pattern with mRNAs. The default settings set in the software were used.

## Overlap with CNV morbidity map (synteny of coordinates, significance calculation) (related to *Figure 1B*)

To find syntenic conservation from mouse to human, UCSC Genome Browser tool Liftover was used. To determine the potential role of lncRNAs in neurodevelopmental disorders, we analyzed array CGH profiles from 29,085 children with intellectual disability and developmental delay that were submitted to Signature Genomics Laboratories, LLC, for clinical microarray-based CGH. The CNV map intersecting lncRNAs was compared with that of 19,584 healthy controls (*Coe et al., 2014*) (dbVar nstd100). Focal enrichment was calculated using fishers exact test statistics and odds ratios comparing cases and control CNV counts at each locus. Validation of focal CNVs affecting lncRNAs of interest was performed on a custom 8-plex Agilent CGH array using standard methods (*Coe et al., 2014*).

## Ingenuity variant analysis (related to *Figure 1B*)

Using Ingenuity Variant Analysis software, we filtered 4,038,671 sequencing variants and obtained a list of 45 variants possibly related with the patient's phenotype. This list included three structural variants (deletions) and one gene fusion. We verified by Sanger sequencing each of the variants associated with disease and found them to be either false positives or non-causative.

The 16.8 Mb large deletion on Chromosome 11 was found to be false positive based on the observation of heterozygosity in the deleted region in whole genome sequencing data. The same false-positive was also shown in the whole genome sequencing data of other two translocation patients. The gene fusion between *ARHGEF3* on Chromosome three and *TRIO* on Chromosome five was determined as false positive by Sanger sequencing. One fragment of *ARHGEF3* (132 bp) intron sequence was inserted into an intron of *TRIO*. The insertion led to the false detection of gene fusion. RT-PCR proved that mRNA splicing of *TRIO* was not affected by this insertion and qRT-PCR proved that the expression level of *TRIO* was not affected. For the rest of variants, we reviewed the reported functions of genes having these variants and copy number variation information in these regions in Database of Genome Variants. We found seven variants occurred in the genes having closely related functions with patient's phenotype and also not extensively covered by CNVs in Database of Genome Variants. We performed Sanger sequencing for these seven variants in patient's family (the father and son having the same translocation and the father also having dyslexia and stutter). Four variants were found to be false positive. Both the patient and healthy mother possessed two variants. All three family members possessed one variant. Overall, we have not found a promising disease causing variants other than the translocation found in patient CMS12200.

## Histone H3 RIP-qRT-PCR (related to *Figure 2F*)

Approximately $20-50 \times 10^6$ cells were used for each experiment. Cells were crosslinked with 1% formaldehyde. Cell pellet was resuspended in equivalent volume of Nuclear Lysis Buffer (i.e. 100mg-1 mL buffer) (50 mM Tris-Cl pH 7.0, 10 mM EDTA, 1% SDS, 100x PMSF, 50x protease inhibitors, and 200x Superase inhibitor). Chromatin was sheared using Covaris sonicator until DNA was fragmented to 200–1000 bp range and diluted 2-fold using Dilution Buffer (0.01% SDS, 1.1% Triton X 100, 1.2 mM EDTA, 16.7 mM Tris-Cl pH 7.0, 167 mM NaCl, 100x PMSF, 50x protease inhibitors, and 200x Superase inhibitor). Samples were incubated with 5 µg of H3 or IgG antibody rotating overnight at 4°C. Protein A dynabeads (50 uL) were washed in Dilution buffer and added to the chromatin for 2 hr rotating at room temperature. Immunoprecipitate fraction was washed four times with Wash buffer (100 mM Tris-Cl pH 7.0, 500 mM LiCl, 1% NP40, 1% sodium deoxycholate, and PMSF). Subsequently, the immunoprecipitate fraction was eluted from beads by vortexing for 30 min at room temperature using elution buffer (50 mM sodium bicarbonate, and 1% SDS). Immediately after 5% of 3M Sodium Acetate was added to neutralize pH. Proteinase K treatment proceeded for 45 min at 45C, followed by RNA extraction using Trizol. Isolated RNA was subjected to DNAse treatment using TurboDNase and purified by phenol-chloroform extraction and ethanol precipitation. For qRT-PCR analysis we used Roche's Lightcycler and Stratagene's RT kit.

## RNA-seq library preparation (related to *Figure 3E and G*, *Figure 3—figure supplement 1C*)

We followed protocols previously described (Wapinski et al., 2013). Briefly, for the RNA-sequencing experiment in *Figures 3* and *4* and S7, libraries were produced from poly-A enriched mRNA using TruSeq kit (Illumina). They were subsequently sequenced using the NextSeq or HiSeq platform producing paired ends reads.

## RNA-seq analysis for loss and gain of function analysis (related to *Figure 3E and G*, *Figure 3—figure supplement 1C*)

Reads obtained were first mapped using Tophat. Expression for each gene was calculated using Cuffdiff (*Figure 3E*, *Figure 3G*) or DEGSeq (*Figure 3—figure supplement 1C*) using default settings. For DEGSeq briefly, only properly paired mapped reads were used (*Wang et al., 2010*). DEGSeq selected longest transcript for each gene, when multiple isoforms were found. Raw counts for each sample were merged into a table and transformed to logarithmic scale. Batch effect among samples was removed using ComBat method in sva package in R (*Leek et al., 2012*). Subsequently, expression values were transformed raw counts and differentially expressed genes were identified by DESeq2 package by comparing different conditions using default parameters (*Leek et al., 2012*). Gene ontology analyses were performed using PANTHER/DAVID.

## ChIRP-seq and data analysis (related to *Figure 4*)

To determine the genome-wide localization of *lnc-Nr2f1*1 we followed protocols previously described (Chu et al., 2011) (*Shen et al., 2012*). ChIRP was performed using biotinylated probes designed against mouse lnc-Nr2f1 using the ChIRP probes designer (Biosearch Technologies). Independent even and odd probe pools were used to ensure lncRNA-specific retrieval (Refer to separate document for odd and even sequences targeting human and mouse lnc-Nr2F1, *Supplementary file 2*). Mouse ES-iN samples are crosslinked in 3% formaldehyde. RNase pre-treated samples are served as negative controls for probe-DNA hybridization. ChIRP libraries are constructed using the NEBNext DNA library preparation kit (New England Biolabs). Sequencing libraries were barcoded using TruSeq adapters and sequenced on HiSeq or NextSeq instruments (Illumina). Reads were processed using the ChIRP-seq pipeline (*Chu et al., 2011*). Even-odd ChIRP-seq tracks are merged as previously described (*Chu et al., 2011*). Peaks were called from the merged tracks over RNase control tracks using MACS14. Overlapping peaks from all replicates were final peaks. High confidence peaks were then filtered by their significance [$-\log 10$ (p-value) $\geq 100$] and correlation between even/odd probes $> 0$, average coverage ($>2$ for mES-iN,$>1$ for hNPC). For hNPC ChIRP of long and short isoforms, the analysis pipeline and filtering criteria are the same. Sequence motifs were discovered using Homer in 200 bp windows. Peak associated gene sets were obtained through GREAT (*McLean et al., 2010*) (http://great.stanford.edu/). Peaks are assigned to genes according to whether peaks are in gene's regulatory domain. Gene regulatory domain is defined as: Each gene is assigned a basal regulatory domain of a distance 5 kb upstream and 1 kb downstream of the TSS (regardless of other nearby genes). The gene regulatory domain is extended in both directions to the nearest gene's basal domain but no more than the 1000 kb extension in one direction. Gene Ontology of gene sets were performed using Metascape (http://metascape.org/). For overlapping mouse ChIRP-seq peak with chromatin features (CTCF, enhancer, H3K27ac, H3K4me3 and PolII) in mouse cortex and E14.5 brain, chromatin annotation files are obtained from public available Chipseq data from Bing Ren lab (*Shen et al., 2012*). For overlapping human ChIRP-seq peak with chromatin features, ChromHMM model of 25 chromatin states and 12 histone modification marks in neuron cells was used (*Ernst and Kellis, 2010*). In addition, peaks are annotated according to distance to genes in *Figure 5* (promoter: $-2$ kb to $+1$ kb of TSS, enhancer: $-2$ kb to $-10$ kb of TSS, exon: exon of a gene, intron: intro of a gene, gene tail: 0 to 2 kb downstream of the end of a gene, intergenic: none of the above).

## Compare ChIP-seq and lncNr2f1 ChIRP-seq

The fastq files of ChIP-seq data were first aligned to mm9 genome using bowtie2 (*Langmead and Salzberg, 2012*). Then the reads with alignment score lower than 10 were removed. The aligned sam files were converted to bam files and sorted by Samtools. Picard (http://broadinstitute.github.

io/picard/) were used to remove duplicates with MarkDuplicates module. After that, Samtools was used to index the bam files. MACS2 was used to call peaks with '-f BAM -g mm -B -p 0.005' options (*Zhang et al., 2008*). Each ChIP-seq peak was annotated by its closest gene using R package 'ChIP-Seeker' (*Yu et al., 2015*), and the number of overlap peaks and genes between mouse ChIRP-seq peaks and ChIP-seq peaks were reported. Random peaks were sampled with the same size of mouse ChIRP-seq peaks and annotated by ChIPSeeker package. The number of overlap peaks and genes between the random peaks and ChIP-seq peaks were recorded. The random sampling procedure was conducted 1000 times to construct null distributions at both peak level and gene level, and the empirical p values were then computed respectively.

## Biochemistry

### Single molecule RNA FISH protocol and probes

Probes were designed using Stellaris probe designer tool and synthesized by Stellaris. Adherent cells were grown in 12 mm coverglass, fixed in 1% formaldehyde for 10 min at room temperature, washed twice with phosphate buffer saline (PBS), and permeabilized using 70% ethanol at 4C overnight. Fixed cells were subjected to RNAse treatment for 30 min at 37C with 0.1 mg/mL RNAse A. After washing (2x SSC, 10% Formamide), hybridization with 250 nM probes in hybridization buffer (10% dextran sulfate, 10% formamide, 2x SSC) at 37C overnight in a coverglass protected from light. The next day, washed (2x SSC, 10% Formamide) at 37C for 30 min. DAPI staining was added to a clean coverslip and coverglass mounted. Slides were images using confocal microscopy.

### In situ hybridization

E13.5 mouse embryos were fixed at 4°C with 4% (weight/volume) paraformaldehyde in PBS overnight. Samples were cryoprotected overnight with 30% (weight/volume) sucrose in PBS, embedded in OCT (Tissue-Tek), and frozen on dry ice. Frozen embryos were sectioned on a cryostat at 16 μm. Sections were processed for in situ hybridization. Frozen sections were treated sequentially with 0.3% (volume/volume) Triton-X in PBS and RIPA buffer (150 mM NaCl, 50 mM Tris-HCl (pH 8.1), 1 mM EDTA, 1% NP-40, 0.5% Sodium Deoxycholate, 0.1% SDS). Sections were postfixed in 4% paraformaldehyde at room temperature for 15 min and washed with PBS. Subsequently, the sections were treated with 0.25% acetic anhydride in 0.1M triethanolamine for 15 min and washed with PBS. Sections were incubated in hybridization buffer (50% formamide deionized, 5 × SSC, 5 × Denhardts, 500 μg/mL Salmon Sperm DNA, 250 μg/mL yeast tRNA) containing DIG-labeled probes at 65°C overnight. Hybridized sections were washed two times in washing solution (2 × SSC, 50% formamide, 0.1% Tween 20) at 65°C for 60 min. After washing, sections were incubated for 1 hr in 1% (weight/volume) blocking reagent (0.1M Maleic Acid, 0.15M NaCl (pH 7.5), 0.1% Tween 20, Roche). Subsequently, incubated with an alkaline phosphatase (AP)-coupled antibody (Roche) at 4°C overnight. After rinsing, the signals were visualized with nitro-blue tetrazolium chloride (NBT)/5-Bromo-4-Chloro-3′-In- dolylphosphatase p-Toluidine salt (BCIP) (Sigma). The DIG-labeled antisense RNA probe for detecting mouse NR2F1 corresponds to the CDS region and for *lnc-Nr2f1* corresponds to 800 bp region. The DIG-labeled sense RNA probe for both, NR2F1 and *lnc-Nr2f1*, corresponds to the same region as the antisense probe in the reverse direction. Probes were generated by in vitro transcription with T7 RNA polymerase (Roche) using the DNA templates containing a promoter sequence of T7 RNA polymerase promoter (TAA TAC GAC TCA CTA TAG GG) followed by a complimentary sequence of target RNA. DNA templates were amplified by PCR with the following primers: For *lnc-Nr2f1* probe: (F) GTG GCC ATG GAA TGG TGT AGC AGA, and (R) GTC TGA GTG TTT GTT TGA CTG AAT GT; NR2F1 probe: (F) CGG TTC AGC GAG GAA GAA TGC CT, and (R) CTA GGA ACA CTG GAT GGA CAT GTA AG.

### Cellular fractionation

Cell fractionation of primary neocortical cells (prepared from E12.5 mouse cerebral cortex) into cytoplasmic and nuclear RNA fractions was performed with a nuclear/cytoplasm fractionation kit (PARIS kit, Ambion) following the instructions of the manufacturer. The chromatin/cytoplasm fractionation was performed following the published protocol (*Conrad and Ørom, 2017*). The amount of RNA in each fraction was determined by qRT-PCR in a Roche LightCycler with Brilliant III Ultra-Fast SYBR Green QRT-PCR Master Mix (Agilent). For primer sequences refer to separate document.

## Immunofluorescence

Cells were fixed with 4% paraformaldehyde for 15 min and subsequently lysed and blocked with blocking buffer [PBS + 0.1% Triton X (Sigma Aldrich)+5% Cosmic calf serum (Thermo Scientific)] for 30 min. Primary antibodies diluted with blocking buffer were added to the wells and left for an hour. The following antibodies were used for immunostaining: mouse anti-MAP2 (Sigma, 1:500), rabbit anti-Tuj1 (Covance, 1:1000), goat anti-Sox1 (R and D, 1:100) and mouse anti-hNestin (R and D, 1:1000). The wells were subsequently washed three times with the blocking buffer. Secondary antibodies conjugated with Alexa dyes (1:1000, Invitrogen) diluted with the blocking buffer were added to the wells and left for an hour. The wells were again washed three times with the blocking buffer. 4',6-Diamidino-2-phenylindole (DAPI) (Life Technologies, 1: 10,000) diluted in PBS was added for 1 min for nuclear staining.

## Western blotting

Cells were lysed with 1 vol of RIPA buffer with cOmplete protease inhibitors (Sigma-Aldrich) and equivolume of 2x Laemmli buffer was added. The samples were then boiled for 5 min at 95°C and subsequently separated in 4–12% Bis-Tris gel with MES buffer (Invitrogen) and transferred onto PVDF membrane for 2 hr at 4°C. Blots were then blocked in blocking buffer (PBS + 0.1% Tween-20 (Sigma-Aldrich) +5% fat-free milk) for 30 min and subsequently incubated overnight with primary antibodies at 4°C. The primary antibodies used are rabbit anti-HSP90 (Cell Signalling) and rabbit anti-NR2F1 (Cell Signalling). The blots were washed three times in PBS + 0.1% Tween-20 for 10 min each. Next, the blots were incubated with secondary antibodies conjugated to horseradish peroxide (Jackson immunoresearch) were diluted in blocking buffer for 1 hr. The blots were washed three times in PBS + 0.1% Tween-20 for 10 min each and once with PBS before adding chemiluminescence substrates (Perkin Elmer) for signal detection on films.

# Acknowledgements

We thank Cindy Skinner, Mrs. Sydney Ladd, Dr. Barbra R DuPont, Dr. Katie R Clarkson for patient recruitment and evaluation and members of our labs for discussion and advice. We thank the Stanford Functional Genomic Facility especially Vanita for her assistant in the project. This project is supported by NIH RC4-NS073015 (HYC, MW), P50-HG007735 (HYC), California Institute for Regenerative Medicine (MW, HYC), NIH R01 HD39331 (AKS) and Self Regional Healthcare Foundation Funds (AKS). CEA was supported by California Institute of Regenerative Medicine Training Grant and Siebel Foundation. QM was supported by Stanford Dean's Fellowship. OLW was supported by a NSF fellowship. MW is a NYSCF–Robertson Stem Cell Investigator. Haplobank is generously funded by Nestlé Institute of Health Science NIHS as well as the Austrian National Bank (OeNB) and Era of Hope/National Coalition against Breast Cancer/DoD. UE is supported by the Austrian Academy of Sciences, the Austrian National Bank (OeNB), and is a Wittgenstein Prize fellow. JMP is supported by an Advanced ERC grant and an Era of Hope/DoD grant. MW is an Howard Hughes Medical Institute Faculty Scholar. HYC and EE are Investigators of the Howard Huges Medical Institute.

# Additional information

## Funding

| Funder | Grant reference number | Author |
| --- | --- | --- |
| NIH Office of the Director | RO1-HD39331 | Anand Srivastava |
| Self Regional Healthcare Foundation | | Anand Srivastava |
| Howard Hughes Medical Institute | | Howard Y Chang Evan E Eichler |
| Howard Hughes Medical Institute | Faculty Scholar | Marius Wernig |

| NIH Office of the Director | RC4-NS073015 | Marius Wernig<br>Howard Y Chang |
| --- | --- | --- |
| California Institute for Regenerative Medicine | | Marius Wernig<br>Howard Y Chang |
| NIH Office of the Director | P50-HG007735 | Howard Y Chang |

The funders had no role in study design, data collection and interpretation, or the decision to submit the work for publication.

## Author contributions

Cheen Euong Ang, Conceptualization, Data curation, Formal analysis, Writing—original draft, Writing—review and editing; Qing Ma, Data curation, Formal analysis, Investigation, Writing—original draft, Writing—review and editing; Orly L Wapinski, Conceptualization, Data curation, Formal analysis; ShengHua Fan, Ryan A Flynn, Bradley Coe, Masahiro Onoguchi, Victor Hipolito Olmos, Investigation; Qian Yi Lee, Brian T Do, Jin Xu, Software; Lynn Dukes-Rimsky, Anand Srivastava, Resources, Investigation; Koji Tanabe, LiangJiang Wang, Ulrich Elling, Josef M Penninger, Evan E Eichler, Resources; Yang Zhao, Formal analysis; Kun Qu, Software, Formal analysis; Marius Wernig, Resources, Supervision, Funding acquisition, Investigation, Writing—review and editing; Howard Y Chang, Conceptualization, Supervision, Funding acquisition, Investigation, Writing—review and editing

## Author ORCIDs

Cheen Euong Ang (iD) http://orcid.org/0000-0002-9050-6122
Qing Ma (iD) http://orcid.org/0000-0001-6812-0584
Qian Yi Lee (iD) http://orcid.org/0000-0001-9200-0910
Jin Xu (iD) http://orcid.org/0000-0003-0944-9835
Josef M Penninger (iD) http://orcid.org/0000-0002-8194-3777
Evan E Eichler (iD) http://orcid.org/0000-0002-8246-4014
Howard Y Chang (iD) http://orcid.org/0000-0002-9459-4393

## Ethics

Human subjects: The study protocol, consent form, consent to publish and privacy practices were reviewed and approved by the Institutional Review Board of the Self Regional Healthcare, Greenwood, SC (Reference number Pro00074882).
Animal experimentation: All mouse work was performed according to IACUC approved protocols at Stanford University (APLAC-21565). Samples in the paper were obtained without determining their sex. All animals were housed in an animal facility with a 12hr light/dark cycle.

## Decision letter and Author response

Decision letter https://doi.org/10.7554/eLife.41770.042
Author response https://doi.org/10.7554/eLife.41770.043

# Additional files

## Supplementary files

• Supplementary file 1. Diagnostic comparison between studies of patients with affected *lnc-NR2F1* locus. Related to *Figure 2* (A) Summary of diagnosis for previously reported patients, including patient CMS12200 described in this study. Highlighted in grey are the shared diagnostic features across patients. Adapted figure (*Al-Kateb et al., 2013*).
DOI: https://doi.org/10.7554/eLife.41770.017

• Supplementary file 2. CHIRP sequencing probes used in the study
DOI: https://doi.org/10.7554/eLife.41770.018

• Supplementary file 3. Public datasets used in the study
DOI: https://doi.org/10.7554/eLife.41770.019

• Supplementary file 4. qRT-PCR primers used in the study

DOI: https://doi.org/10.7554/eLife.41770.020

• Supplementary file 5. RNA FISH primers used in the study
DOI: https://doi.org/10.7554/eLife.41770.021

• Supplementary file 6. Sequence conservation used in the study
DOI: https://doi.org/10.7554/eLife.41770.022

• Supplementary file 7. A list of human lncRNAs reported in the study
DOI: https://doi.org/10.7554/eLife.41770.023

• Supplementary file 8. A list of mouse lncRNAs reported used in the study
DOI: https://doi.org/10.7554/eLife.41770.024

• Transparent reporting form
DOI: https://doi.org/10.7554/eLife.41770.025

## Data availability

A summary table containing all the lnc-Nr2f1 mutation reported in the literature (Supplementary file 1), the ChIRP-sequencing probes (Supplementary file 2), datasets used in this paper and their corresponding source (Supplementary file 3), the qRT-PCR primer sequences (Supplementary file 4), RNA FISH (Supplementary file 5) and the sequence conservation (Supplementary file 6) can be found in the supplementary documents. The sequence of human and mouse lncRNAs reported in paper are in the Supplementary file 7 and Supplementary file 8 respectively. The datasets generated during and/or analyzed during the current study are available in the NIH GEO repository (GSE115079).

The following dataset was generated:

| Author(s) | Year | Dataset title | Dataset URL | Database and Identifier |
|---|---|---|---|---|
| Ang CE, Ma Q, Wapinski OL, Fan S | 2018 | Sequencing data from The novel lncRNA lnc-NR2F1 is pro-neurogenic and mutated in human neurodevelopmental disorders | https://www.ncbi.nlm.nih.gov/geo/query/acc.cgi?acc=GSE115079 | NCBI Gene Expression Omnibus, GSE115079 |

The following previously published datasets were used:

| Author(s) | Year | Dataset title | Dataset URL | Database and Identifier |
|---|---|---|---|---|
| Ayoub AE, Oh S, Xie Y, Leng J, Cotney J, Dominguez MH, Noonan JP, Rakic P | 2011 | Transcriptional programs in transient embryonic zones of the cerebral cortex defined by high-resolution mRNA-sequencing | https://www.ncbi.nlm.nih.gov/geo/query/acc.cgi?acc=GSE30765 | NCBI Gene Expression Omnibus, GSE30765 |
| Gregg C, Dulac C | 2010 | High resolution analysis of genomic imprinting in the embryonic and adult mouse brain AND Sex-specific imprinting in the mouse brain | https://www.ncbi.nlm.nih.gov/geo/query/acc.cgi?acc=GSE22131 | NCBI Gene Expression Omnibus, GSE22131 |
| Fietz SA, Huttner WB, Pääbo S | 2012 | Transcriptomes of germinal zones of human and mouse fetal neocortex suggest a role of extracellular matrix in progenitor self-renewal. | https://www.ncbi.nlm.nih.gov/geo/query/acc.cgi?acc=GSE38805 | NCBI Gene Expression Omnibus, GSE38805 |
| Belgard TG, Marques AC, Oliver PL, Ozel Abaan H, Sirey TM, Garcia-Moreno F, Molnar Z, Margulies EH, Ponting CP | 2011 | A Transcriptomic Atlas of Mouse Neocortical Layers | https://www.ncbi.nlm.nih.gov/geo/query/acc.cgi?acc=GSE27243 | NCBI Gene Expression Omnibus, GSE27243 |
| Ramos A, Nellore A | 2013 | Integration of genome-wide approaches identifies lncRNAs of adult neural stem cells and their progeny in vivo | https://www.ncbi.nlm.nih.gov/geo/query/acc.cgi?acc=GSE45282 | NCBI Gene Expression Omnibus, GSE45282 |

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
