## [Decision Letter]

Thank you for submitting your article "The novel lncRNA *lnc-NR2F1* is pro-neurogenic and mutated in human neurodevelopmental disorders" for consideration by *eLife*. Your article has been reviewed by three peer reviewers, one of whom is a member of our Board of Reviewing Editors, and the evaluation has been overseen by a Reviewing Editor and Marianne Bronner as the Senior Editor. The reviewers have opted to remain anonymous.

The reviewers have discussed the reviews with one another and the Reviewing Editor has drafted this decision to help you prepare a revised submission.

Summary:

This study characterizes the gene regulatory functions of *lnc-NR2F1* in neuronal cell maturation and its potential implication in ASD/ID. Previously the lncRNAs have been shown to have gene regulatory functions in various contexts and even implicated in diseases through genome-wide correlation studies. Nonetheless, scant information has been available for demonstrating a causal role of lncRNAs in the pathogenesis of a disease.

This study opens with a differential screen for regulated lncRNA expression in induced neurons reprogrammed from MEFs by BAM factor expression. This allows the authors to focus in on the set of lncRNAs that are most likely to be relevant in neural differentiation. They then map these lncRNAs to human homologs and look for cases in which they fall within regions that have neural disease associated CNVs. Validation of a small number of these lncRNAs confirms their altered expression in ASD/ID patient-derived cells. One of these, *lnc-NR2F1*, is within a larger region associated with developmental disability, but the authors map at least one lncNR2F1-restricted deletion in a child and father. To test the function of this lncRNA in neural development they generated matched sets of mouse ES cells with overexpression or loss-of-function and demonstrate cellular effects which, though somewhat subtle, seem consistent. Finally, the authors provide a series of experiments in which they use the lncRNA as a substrate for DNA IP to find the bound targets, then compare against gene expression data for the validation of target regulation. These data tell a compelling story about *lnc-NR2F1* as a contributor to the gene expression program that controls neuronal differentiation.

The manuscript has two significant strengths – one is the overall strategy, which as the authors suggest could be used as a broader platform for identification of additional neural development-associated lncRNAs. The second is the biochemical study of lncNR2F1, which offers a novel characterization of this lncRNA. The study does not go very far towards establishing the specific biological functions of this lncRNA, though the cellular assays combined with the genetics are certainly strongly suggestive that it plays an important role, and further functional studies were viewed by the reviewers as beyond the scope of the current study. However, we did have several questions/comments mainly related to the studies about the evolutionarily conserved features of the lncRNA and the isoform-specific differences in the functionality, as well as some clarifications/controls that we considered essential to address, some of which may require additional experimentation.

Essential revisions:

1) The reviewers had several questions/concerns about the experiments regarding the distinct isoforms of *lnc-Nr2f1* that need to be addressed either through experimentation or clarification. These include the following:

The authors emphasized the strong conservation in exon sequences among different species and isoform-specific functionality in gene regulation and disease-relevance. Indeed, evidence was presented to show that the short isoform of the human *lnc-Nr2f1* is nonfunctional in the neuronal maturation process and disease manifestation. But all isoforms contain the three conserved exons, and it appears that the functional mouse *lnc-Nr2f1* is structurally most similar to the human short isoform. It would be more compelling if a structure-function analysis of *lnc-Nr2f1* could be performed to determine the critical functional domain of the lncRNA in gene regulation. It could be that a specific domain structure or sequence in within the *lnc-Nr2f1* commonly mediates the interactions with bHLH factors and others.

There are two isoforms of *lnc-Nr2f1* present in the mouse. Please describe which isoform was used or targeted in the experiments or whether either isoform was equally effective or not. Only the shorter isoform contains all three conserved exons, thus the isoform-specific analysis could provide functional evidence for the importance of the conserved sequences.

The presence of all human/mouse isoforms with expected size should be experimentally validated (e.g., Northern).

In Figure 5B and Figure 5—figure supplement 1F, it should be shown that the levels of overexpressed isoforms are comparable to each other to ensure that the functional differences are not caused by the expression level difference. Figure 5B does not look convincing in that the quantitative differences between the isoforms 2 and 3 are very small, which could be due to the quantitative differences in the expression levels of individual isoforms. In Figure 5B, the two presumably functional isoforms 1 and 2 results in the opposite phenotypes in the maturation assay, which is a bit hard to interpret.

2) The following point needs to be resolved: Figure 2D and E do not seem to match. RNA FISH analysis shows a predominant localization in the nucleus whereas cellular fractionation shows a slightly more enrichment in the cytoplasm. Was the RNA FISH probe designed from the exonal regions only, which will detect only mature form? It would be important to experimentally validate whether *lnc-Nr2f1* is indeed spliced and resides in the nucleus, and also to quantify the relative abundance of the mature and precursor forms present in the developing neurons. It might be that the mature form will leave the nucleus and localized in the cytoplasm.

3) We suggest strengthening the data to address the following concern: Given the strong sequence conservations in exons among different species, it would be advised to analyze the potential protein-coding capacity of *lnc-Nr2f1*. The protein-coding capacity analysis could be done without much work and even computationally, but it is important to make sure that the lncRNA acts as RNA.

4) Finally, two points were raised that if addressed would help to strengthen the story, though we did not see them as essential for acceptance of the manuscript. We include these concerns here for the authors in case they have data or clarifications that they may wish to add in order to resolve these points:

The first group of questions related to the cellular phenotype of the lncRNA KO: Is the *lnc-Nr2f1* KO phenotype consistent with a reduction in the number and neurite length of neurons? It would be of interest to know whether there is a direct effect of the knockout of *lnc-Nr2f1* on the Ascl1-mediated neuronal reprogramming of MEFs. The rescue experiment in ES cells nicely demonstrates the specificity of *lnc-Nr2f1* LOF analysis, and further implies that *lnc-Nr2f1* might work in trans. But the authors did not examine the neuronal maturation phenotypes of the *lnc-Nr2f1* LOF. For comparison, *Nr2f1* KO did not result in any change in the neurite length or number in mES-iN cells which led the authors to conclude that *lnc-Nr2f1* might function independently of the neighboring gene. It would be more complete evidence if a phenotypic consequence from *lnc-Nr2f1* LOF is observed. The loss-of-function study in ES cells should be accompanied by an experimental validation of the mutagenesis cassette-dependent loss of *lnc-Nr2f1*.

The second set of questions relates to the transcriptional mechanism of the lncRNA action: An unresolved question is how *lnc-Nr2f1* promotes neurogenic genes. The ChIRP analysis suggested a model shown in Figure 4F that *lnc-Nr2f1* binds to the bHLH motif-enriched sites to regulate the downstream neuronal genes. One possible scenario is that it could cooperate with bHLH neurogenenic factors to facilitate their functions in gene expression. Assuming that *lnc-Nr2f1* itself is not sufficient for promoting neuronal reprogramming (or is it?), the *lnc-Nr2f1* might require to work with overexpressed Ngn2 in ES cells in the model system. If then, shouldn't its ChIRP peak sites be significantly overlapped with Ngn2 motifs? Alternatively, ChIP-seq of overexpressed Ngn2 could be performed to examine the binding site overlap between Ngn2 and *lnc-Nr2f1*.

---

## [Author Response]

Essential revisions:1) The reviewers had several questions/concerns about the experiments regarding the distinct isoforms of lnc-Nr2f1 that need to be addressed either through experimentation or clarification. These include the following:The authors emphasized the strong conservation in exon sequences among different species and isoform-specific functionality in gene regulation and disease-relevance. Indeed, evidence was presented to show that the short isoform of the human lnc-Nr2f1 is nonfunctional in the neuronal maturation process and disease manifestation. But all isoforms contain the three conserved exons, and it appears that the functional mouse lnc-Nr2f1 is structurally most similar to the human short isoform. It would be more compelling if a structure-function analysis of lnc-Nr2f1 could be performed to determine the critical functional domain of the lncRNA in gene regulation. It could be that a specific domain structure or sequence in within the lnc-Nr2f1 commonly mediates the interactions with bHLH factors and others.

As reported, the two mouse *lnc-Nr2f1* isoforms and the three human *lnc-Nr2f1* isoforms share three highly conserved exons. However, the last exon of both short or long mouse *lnc-Nr2f1* vs. human is not conserved. It is thus difficult to draw a parallel between the mouse *lnc-Nr2f1* and the short isoform of human *lnc-Nr2f1*. The other pertinent point is all three isoforms we used in the paper are about the same length: 3620, 2697, 2814bp for isoform I, II and III respectively. We used long and short to imply if it spans a long or short genomic loci. To address the reviewer question, we would like to point the data we showed in the original figures.

We performed domain ChIRP using probes targeting the short and long isoform-specific exons (Figure 5A). Comparing to the long isoform of human *lnc-Nr2f1*, the short isoform of human *lnc-Nr2f1* binds to significantly fewer genomic loci (Figure 5C) and the genes in cis are not enriched for any neuronal gene ontology terms. In contrast, the genomic targets of the long isoform of *lnc-Nr2f1* is enriched for neuronal GO term) (Figure 5F).

We overexpressed each of the three isoforms of human *lnc-Nr2f1* in neuroblastoma cell lines, and only the first and second long isoform gave significant up/downregulation of a set of genes. The third and shorter isoform did not change the gene expression (Figure 5—figure supplement 1F). These results suggest that the sequences present in the long isoform are most relevant to *lnc-Nr2f1* function described in this work.

There are two isoforms of lnc-Nr2f1 present in the mouse. Please describe which isoform was used or targeted in the experiments or whether either isoform was equally effective or not. Only the shorter isoform contains all three conserved exons, thus the isoform-specific analysis could provide functional evidence for the importance of the conserved sequences.

For the mouse experiments, we used a single isoform of mouse of *lnc-Nr2f1* (named A830082K12Rik, sequence outlined in the supplementary document) because only this transcript was upregulated during iN reprogramming (Figure 1—figure supplement 1H). During iN reprogramming, we only detected the expression of this isoform (see red boxes in UCSC browser tracks Author response image 1). The first exon of the other mouse *lnc-Nr2f1* isoforms (named AK044036 and AK051417, see green box in UCSC browser track) has no detected read in both the BAM 22d sample and cortical neurons.

The presence of all human/mouse isoforms with expected size should be experimentally validated (e.g., Northern).

We attempted Northern analysis twice but was not successful. We observed a smear in the Northern and saw no difference in the hybridization signal for WT and *lnc-Nr2f1* KO cells. Hence, we believe that the Northern signal was nonspecific, and thus far is not informative of RNA isoforms. We therefore turned to a different strategy to validate the RNA isoform annotations that was used. We interrogated the published long RNA-sequencing data of brain cells from Tilgner and coworkers (Tilgner, 2013), we found that all isoforms that we reported in the paper were detected in vivo.

**Author response image 2. respfig2:** 

In Figure 5B and Figure 5—figure supplement 1F, it should be shown that the levels of overexpressed isoforms are comparable to each other to ensure that the functional differences are not caused by the expression level difference. Figure 5B does not look convincing in that the quantitative differences between the isoforms 2 and 3 are very small, which could be due to the quantitative differences in the expression levels of individual isoforms. In Figure 5B, the two presumably functional isoforms 1 and 2 results in the opposite phenotypes in the maturation assay, which is a bit hard to interpret.

The experiments were performed into two distinct cell types. Figure 5B shows that overexpression of the three isoforms of human *lnc-NR2F1* improve the maturation of mouse induced neuronal cells, while Figure 5—figure supplement 1F shows the transcriptional effects of overexpressing the three isoforms of human *lnc-NR2F1* in human neuroblastoma cells. One can reasonably conclude that overexpression of isoform 2 leads to better iN maturation compared to isoform 3 (Figure 5B) and this is likely due to the differences in transcriptional responses (note the number of differentially expressed genes in Figure 5—figure supplement 1F). We agree that it was unexpected to find isoform 1, while giving the highest transcriptional responses, hampers iN reprogramming. We know that cell fate reprogramming is a process involving sequential dynamic waves of chromatin changes and gene expression. There might be a precarious balance where too much gene expression or expression ahead of schedule reduces productive neuronal maturation.

2) The following point needs to be resolved: Figure 2D and E do not seem to match. RNA FISH analysis shows a predominant localization in the nucleus whereas cellular fractionation shows a slightly more enrichment in the cytoplasm. Was the RNA FISH probe designed from the exonal regions only, which will detect only mature form? It would be important to experimentally validate whether lnc-Nr2f1 is indeed spliced and resides in the nucleus, and also to quantify the relative abundance of the mature and precursor forms present in the developing neurons. It might be that the mature form will leave the nucleus and localized in the cytoplasm.

Thank you for pointing out this point that benefits from further clarification. RNA FISH showed predominant enrichment in the nucleus, but there were also detectable signals in the cytoplasm. We have replaced Figure 2D with a longer exposure of the RNA FISH to show the cytoplasmic signal. Thus, the cellular fractionation and the RNA FISH agrees with each other.

The probes in the RNA FISH target only the exons; therefore, the strong nuclear signal of *lnc-Nr2f1* is not due to intronic signal from nascent RNAs. To determine whether mouse *lnc-Nr2f1* is indeed spliced, we performed fractionation followed by RT-PCR on day 4 mouse embryonic stem cells derived induced neurons. As one can see from Author response image 3, spliced mouse *lnc-Nr2f1* resides both in the nucleus (especially in the chromatin) and in the cytoplasm as indicated by two independent primer pairs that span exon-exon junction. These multiple lines of evidence suggest that the predominant pool (if not all) of mature *lnc-Nr2f1* is nuclear and associated with chromatin.

**Author response image 3. respfig3:** (**A**) Gel showing the different RNA species in different cellular fractions. The protocol reported by Conrad et al., 2017 was used. (**B**) Gel showing the RT-PCR using cytoplasmic or chromatin fractions using GAPDH (positive control), Xist (nuclear localized), two independent exon-exon and exon-intron primers.

3) We suggest strengthening the data to address the following concern: Given the strong sequence conservations in exons among different species, it would be advised to analyze the potential protein-coding capacity of lnc-Nr2f1. The protein-coding capacity analysis could be done without much work and even computationally, but it is important to make sure that the lncRNA acts as RNA.

Thank you for this excellent suggestion. We analyzed the all mouse and human *lnc-Nr2f1* sequences using the coding Potential Calculator (Kong et al., 2007), and obtained a negative score for all of them which implies that they are unlikely to code for proteins. We added the table in the supplementary table.

**ID****C/NC****CODING POTENTIAL SCORE**A830082K12Rik/NR_045195 (mouse *lnc-Nr2f1*)noncoding-0.40NR_109825.1(human *lnc-Nr2f1* I)noncoding-1.02066NR_021491.2 (human *lnc-Nr2f1* II)noncoding-0.997853NR_021490.2 (human *lnc-Nr2f1* III)noncoding-0.85903

4) Finally, two points were raised that if addressed would help to strengthen the story, though we did not see them as essential for acceptance of the manuscript. We include these concerns here for the authors in case they have data or clarifications that they may wish to add in order to resolve these points:The first group of questions related to the cellular phenotype of the lncRNA KO: Is the lnc-Nr2f1 KO phenotype consistent with a reduction in the number and neurite length of neurons? It would be of interest to know whether there is a direct effect of the knockout of lnc-Nr2f1 on the Ascl1-mediated neuronal reprogramming of MEFs. The rescue experiment in ES cells nicely demonstrates the specificity of lnc-Nr2f1 LOF analysis, and further implies that lnc-Nr2f1 might work in trans. But the authors did not examine the neuronal maturation phenotypes of the lnc-Nr2f1 LOF. For comparison, Nr2f1 KO did not result in any change in the neurite length or number in mES-iN cells which led the authors to conclude that lnc-Nr2f1 might function independently of the neighboring gene. It would be more complete evidence if a phenotypic consequence from lnc-Nr2f1 LOF is observed. The loss-of-function study in ES cells should be accompanied by an experimental validation of the mutagenesis cassette-dependent loss of lnc-Nr2f1.

In the *lnc-Nr2f1* knockout, we did not observe significant phenotypes in the number or length of neurite, despite the strong downregulation of axonal pathfinding genes. The down regulation of axon pathfinding genes in *lnc-Nr2f1* KO was rescued when we expressed lnc-Nr2f1 in trans. The axon pathfinding genes raise the possibility that axonal pathfinding or neuronal circuit connection may be affected. These additional studies are best addressed using in vivo model and might be beyond the scope of the present study but will be of interest to explore in the future.

The second set of questions relates to the transcriptional mechanism of the lncRNA action: An unresolved question is how lnc-Nr2f1 promotes neurogenic genes. The ChIRP analysis suggested a model shown in Figure 4F that lnc-Nr2f1 binds to the bHLH motif-enriched sites to regulate the downstream neuronal genes. One possible scenario is that it could cooperate with bHLH neurogenenic factors to facilitate their functions in gene expression. Assuming that lnc-Nr2f1 itself is not sufficient for promoting neuronal reprogramming (or is it?), the lnc-Nr2f1 might require to work with overexpressed Ngn2 in ES cells in the model system. If then, shouldn't its ChIRP peak sites be significantly overlapped with Ngn2 motifs? Alternatively, ChIP-seq of overexpressed Ngn2 could be performed to examine the binding site overlap between Ngn2 and lnc-Nr2f1.

We thank the reviewers’ helpful suggestions to potential mechanism of *lnc-Nr2f*1. We performed and analyzed ChIP-seq data of bHLH neurogenic factors Ngn2 and Ascl1 as suggested. We then compared the lncNr2f1 ChIRP-seq data with the ChIP-seq data at the level of individual peaks and peak associated genes (nominated by the algorithm GREAT). We found that direct overlap between lncNr2f1 ChIRP-seq peaks and ChIP-seq peaks are low but statistically significant. 3.2% and 13.2% lncNr2f1 ChIRP-seq peaks overlapped with or binding within 5kb to Ngn2 and Ascl1 ChIP peaks, respectively (Figure 4—figure supplement 1F,G). Despite that, lncNr2f1 target genes overlap very significantly with Ngn2 (37.9%) or Ascl1 (67.3%) target genes (Figure 4—figure supplement 1F,G) at the level of peak-associated genes. These results suggest that lncNr2f1 and bHLH transcription factors such as Ascl1 and Ngn2 may coordinately regulate the same set of neuronal genes, and the majority of instances occur with *lnc-Nr2f1* and the bHLH factors binding nearby but non-overlapping sites. We have added a sentence in the Results section to report this finding.